# Engineered helicase replaces thermocycler in DNA amplification while retaining desired PCR characteristics

Momčilo Gavrilov [1], Joshua Y. C. Yang[2], Roger S. Zou [2], Wen Ma[3], Chun-Ying Lee [4], Sonisilpa Mohapatra[1], Jimin Kang[4], Ting-Wei Liao[4], Sua Myong [4] & Taekjip Ha [1,2,4,5]

Polymerase Chain Reaction (PCR) is an essential method in molecular diagnostics and life sciences. PCR requires thermal cycling for heating the DNA for strand separation and cooling it for replication. The process uses a specialized hardware and exposes biomolecules to temperatures above 95 °C. Here, we engineer a PcrA M6 helicase with enhanced speed and processivity to replace the heating step by enzymatic DNA unwinding while retaining desired PCR characteristics. We name this isothermal amplification method SHARP (SSB-Helicase Assisted Rapid PCR) because it uses the engineered helicase and single-stranded DNA binding protein (SSB) in addition to standard PCR reagents. SHARP can generate amplicons with lengths of up to 6000 base pairs. SHARP can produce functional DNA, a plasmid that imparts cells with antibiotic resistance, and can amplify specific fragments from genomic DNA of human cells. We further use SHARP to assess the outcome of CRISPR-Cas9 editing at endogenous genomic sites.

Nucleic acids amplification is an essential method involved in rapid molecular diagnostics, gene manipulations, and genetic analyses, including the detection of bacteria, viruses, and diagnosis of genetic disorders. The most widely used method for DNA amplification is Polymerase Chain Reaction (PCR), implemented through thermal cycling. Promising next-generation methods include enzymatic isothermal amplification; however, existing isothermal methods are limited to producing only short amplicons, or complex heterogeneous mixed and branched products, and often require multiple sets of complicated pairs of primers[1,2]. As a result, to the best of our knowledge, no existing isothermal method can approach the versatility of thermocycler-based PCR.

There are more than 10 different approaches for isothermal amplification[1–3] which differ in how they initiate each reaction cycle and allow primers to bind the template. PCR heat denatures the DNA duplex and subsequently cools it down to initiate each replication cycle (Fig. 1a), while isothermal methods either use specially designed primers or other assisting proteins to enable primer binding to the template. LAMP (Loop-Mediated Isothermal Amplification) uses 4 to 6 loop-forming primers to generate 3′ ends that are extendable by Bst DNA polymerase (DNAP)[4]. LAMP is a rapid, commercially available[5], and highly sensitive exponential amplification method. However, LAMP products are highly complex, hard to interpret, compare, and use in applications such as cloning, gene manipulation, and sequencing. Furthermore, LAMP primer design requires specialized software, and primers optimized for PCR cannot be used as LAMP primers. Isothermal amplification methods based on gp32, a single-stranded DNA binding protein, uses a set of two primers and can produce over 1 kilo base-pair (bp) amplicons[3]; however, the template is amplified only a few times, far from the exponential growth in PCR, and the product also contains high molecular-weight multimers and other undesired bands. RPA (Recombinase Polymerase Amplification) uses two primers

[1]Department of Biophysics and Biophysical Chemistry, Johns Hopkins University School of Medicine, Baltimore, MD, USA. [2]Department of Biomedical Engineering, Johns Hopkins University School of Medicine, Baltimore, MD, USA. [3]Department of Chemistry and Biochemistry, University of California, San Diego, CA, USA. [4]Department of Biophysics, Johns Hopkins University, Baltimore, MD, USA. [5]Howard Hughes Medical Institute, Baltimore, MD, USA. ✉e-mail: tjha@jhu.edu

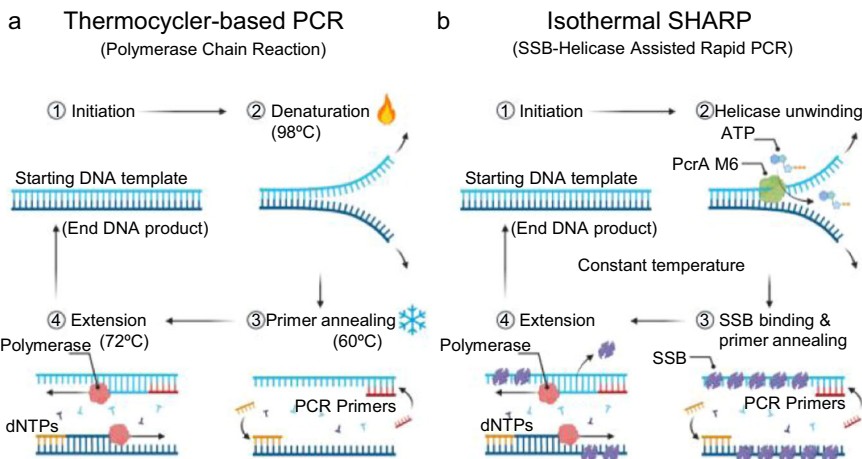

**Fig. 1 | Comparison of PCR and SHARP. a** PCR alternates the temperature between high (≈98 °C) to melt the DNA duplex and low (45–65 °C) to replicate the complementary strand by a DNA polymerase. **b** SHARP uses PcrA M6 helicase and SSB to open the duplex and enable primer binding, followed by Bst-LF polymerase to replicate the strand. SHARP occurs at a constant temperature. Created with BioRender.com.

and a recombinase[6,7]. RPA primers are sometimes longer than PCR primers, 30 to 38 bases, which is necessary for recombinase binding. The recombinase-primer complex then searches for homologous template sequence resulting in strand invasion and primer-template pairing, followed by polymerase extension. RPA provides exponential amplification and a competitive detection limit, but it can produce only short amplicons, 100 to 200 bp in length. Longer primers used by RPA are more prone to forming secondary structure and nonspecific product, requiring more careful primer design, or unnatural bases[8]. SIBA (Strand Invasion-Based Amplification) also uses a recombinase, UvsX, but it reduces the possibility of the undesired product formation by using an invasion oligonucleotide in addition to primers[9,10] but still generates only short amplicons. SDA (Strand Displacement Amplification)[11] and NEAR (Nicking Enzyme Amplification Reaction)[12] are similar methods that use a nicking enzyme to assist the amplification initiation. Both methods provide exponential amplification, but the product is also only several hundred base pairs long. Both methods use a set of 4 primers, with additional sequence requirements for the nicking enzyme.

HDA, or Helicase-Dependent Amplification[13,14], utilizes a DNA helicase to generate single-stranded templates for primer hybridization and subsequent extension by a DNAP. HDA is commercially available, and it uses similar primers to PCR. Combining UvrD helicase, an accessory protein MutL that enhances UvrD activity[15], T4 gene 32 protein (SSB), and Klenow Exo⁻ DNAP, Vincent et al. achieved an exponential amplification of 100 bp product[13]; however, for longer amplicons the reaction yield dropped significantly, limiting the possible applications of the method. Thermophilic HDA (tHDA)[16], implemented with Tte-UvrD, simplified the reaction components by excluding MutL and SSB. However, tHDA kits have the same limitation as HDA because they can amplify and detect only short DNA sequences (70 bp–120 bp) and use primers concentrations below 75 nM, limiting the product yield. Because of these limitations, commercial tHDA has been utilized only for several diagnostic tests[14]. The general idea of using a helicase instead of heating for splitting the DNA was proposed in the original patent for PCR[17], but strand separation by helicases has not been widely utilized for DNA amplification which we attribute to the lack of helicases suitable for this application.

In this work we engineer PcrA M6 helicase to replace a thermocycler in DNA amplification while retaining the versatility and other desired characteristics of PCR. We explore the mechanistic properties of the engineered helicase with Single-molecule Picometer Resolution Nanopore Tweezers (SPRNT) and molecular dynamic simulations, and we show that the enhanced processivity and speed of the engineered PcrA M6 helicase facilitate isothermal amplification.

## Results and discussion

We found that an engineered superhelicase that can unwind thousands of base pairs of DNA progressively[18] allows amplification of DNA much longer than was possible using conventional helicases. We initially developed a protocol using the PcrA-X superhelicase where a thermophilic *Geobacillus stearothermophilus* PcrA helicase is constrained into the unwinding-active conformation through crosslinking two cysteines inserted into the protein[18]. Later we found that applying crosslinker molecule to our PcrA M5 helicase is not necessary, because the two introduced cysteine residues are 4A apart in the unwinding active, closed conformation and their position enables the disulfide bond formation without an external crosslinker, thus confining PcrA M5 to the unwinding active conformation. We also made one additional mutation (PcrA M6) to increase the helicase unwinding activity and speed up to 4 times. With PcrA M6 helicase, we present a robust isothermal amplification method analogous to PCR in that it uses the same set of primers and template as PCR at the input and outputs the same amplicon as PCR, with the length of up to 6000 base pairs (bp). We named our amplification method SHARP (SSB-Helicase Assisted Rapid Polymerase chain reaction) because it uses engineered helicase and SSB (Single-Stranded DNA Binding protein) in addition to the standard PCR reagents.

SHARP retains all desired PCR characteristics in place whilst adding additional features. SHARP performs equally well or better than PCR according to 8 criteria we benchmarked against: (1) up to 6000 bp amplicon length, (2) 5–30 min amplification time, (3) primer design principles and convenience, (4) detection limit, (5) real-time detection, (6) amplification-result interpretation, (7) downstream product applications, and (8) no initial heat-denaturing step. Numerous benefits of avoiding thermal cycling from PCR for faster diagnostics are often discussed in the literature[1–3], and here we also test the suitability of isothermal reaction for other tasks in the wet lab. SHARP can quantify the level of CRISPR-Cas9 genome editing inside cells[19]. *E. coli* cells transformed with 3.2 kbp SHARP-made plasmid can replicate the plasmid and display antibiotic resistance. DNA sequences with potential to form non-canonical structures, such as sequence containing (CAG)₄₇ repeats, do not inhibit SHARP. SHARP can be carried out in the wide temperature range between 37 and 65 °C, leaving many possibilities for future optimizations, combination with other techniques, and usage with primers and molecular probes of different annealing temperatures.

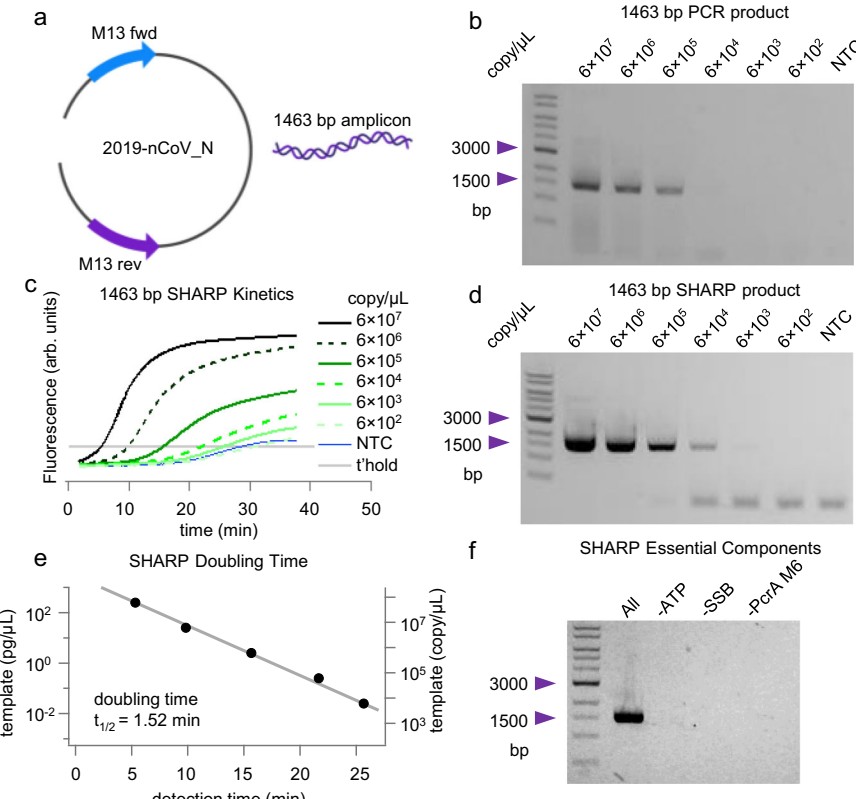

**Fig. 2 | PCR and SHARP amplification product. a** 2019 coronavirus nCoV-2 N protein sequence between M13 primers. **b** PCR product and detection limit for the template-primer set in **a**. **c** Kinetics of the SHARP reaction for the template primer set in a and different template concentrations. NTC is no template control and t'hold is the chosen detection threshold. **d** SHARP product and detection limit enables the direct comparison with PCR. **e** SHARP doubling time estimate. Detection time is the intercept of the intensity curve with the threshold in **c**. **f** SHARP essential components. In the absence of ATP, SSB, or PcrA M6 no amplification of the 1463 bp product occurs. The standard deviation of the template copy number is 4%, see Methods for details. Source data are provided as a Source Data file.

## Reference PCR assay

To test SHARP's ability to perform according to the 8 criteria presented above, we used a linearized 4012-bp DNA template vector containing 2019 coronavirus nCoV-2 N protein sequence between M13 primers (Fig. 2a). Fig. 1a summarizes the thermocycler-based PCR as the reference reaction[20]. We mix nCoV-2 N template, two M13 primers at 250 nM each with the commercial NEB (New England Biolabs, #M0531S) Phusion master mix containing PCR components. The thermocycler alternates the temperature between 98 °C to melt the duplex, 65 °C for primers to bind, and 72 °C to extend primers bound to the template. After 30 cycles, each template molecule is theoretically copied $2^{30} \approx 10^9$ times, leading to a macroscopically detectable 1463 bp product. From the amounts of single-band PCR amplicon in 2% agarose gel vs. the initial template copy number in the range from $6 \times 10^2$ to $6 \times 10^7$ per microliter (Fig. 2b), we determine that the reference PCR needs at least $6 \times 10^3$ copies of the template for gel-based detection of the 1463 bp product.

## SHARP amplification

We use SHARP to carry out the amplification reaction at 65 °C, with the same template copy numbers and the same primer set as the reference PCR. SHARP produces the same 1463 bp amplicon as PCR (Fig. 2d). We also monitored DNA amplification in real time using intercalating dyes (Fig. 2c). Principal components of SHARP are 1 template, 2 primers at 250 nM each, dNTPs, Bst-LF DNAP, *E coli* SSB, engineered PcrA M6 helicase (see the section below titled Engineering PcrA helicase for SHARP for its characterization), ATP, and thermostable inorganic pyrophosphatase, see Table 1. Excluding SSB, PcrA M6 or ATP prevents the amplification (Fig. 2f). In the development phase, we also tested other polymerases (the large fragment of Bsu, Klenow Exo⁻, phi29, and reverse transcriptase M-MLV and AMV). They all showed some degree of amplification, but none of these other polymerases satisfied all 8 criteria. Later in the article we show details on the amplification results with Bsu and Klenow Exo- at 37 °C and the effect of SSB concentration on the product yield.

Figure 2c shows SHARP kinetics as a function of the initial template copy number. For $6 \times 10^7$ copies per microliter, the SHARP product is detectable after 5 min, while for $6 \times 10^3$ it takes about 22 min at 65 °C. The 1463 bp SHARP product imaged on 2% agarose gel in Fig. 2d reproduces the same bands observed for the reference PCR product (Fig. 2b). The detection limit of SHARP is above $6 \times 10^2$ copies per microliter and more sensitive than the detection limit of $6 \times 10^3$ for PCR in Fig. 2b. Comparing SHARP (Fig. 2d) and PCR (Fig. 2b) products

## Table 1 | SHARP components for 40 µL reaction volume

| Component 1 (2X): | µL | Component 2 (2X): | µL |
| --- | --- | --- | --- |
| Nuclease free water | 9 | Nuclease free water | 11.1 |
| 10X Buffer | 2 | 10X Buffer | 2 |
| EvaGreen (20X) | 2 | DTT (100 mM) | 4 |
| dNTPs (10 mM each) | 2 | Bst-LF (1.5 mg mL⁻¹) | 0.2 |
| ATP (100 mM) | 2 | SSB (9 mg mL⁻¹) | 0.5 |
| Forward primer (10 uM) | 1 | PcrA M6 (0.2 mg mL⁻¹) | 2 |
| Reverse primer (10 uM) | 1 | Thermostable PPase (100 units mL⁻¹) | 0.2 |
| Template (variable conc.) | 1 | | |
| Total | 20 | Total | 20 |

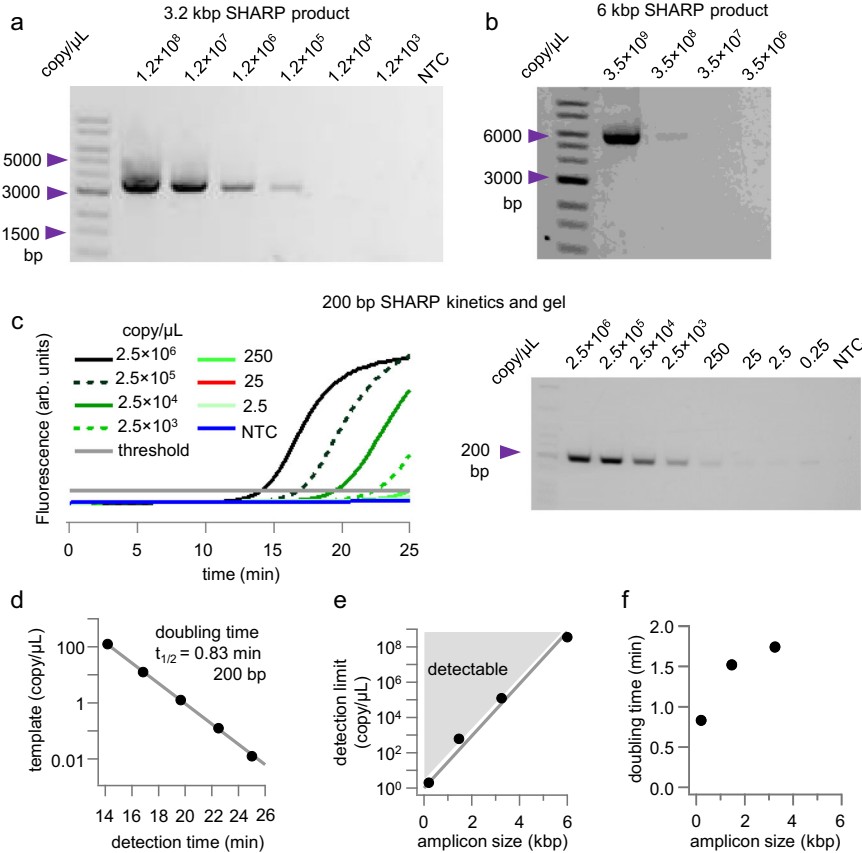

**Fig. 3 | Other SHARP products. a** 3 kbp SHARP amplicon. **b** 6 kbp SHARP amplicon. **c** 200 bp SHAR amplicon using λDNA as a template. **d** SHARP doubling time estimate for 200 bp product. **e** Detection limit versus the amplicon length. The area above the detection limit is shaded in gray and labeled detectable. **f** Doubling time shows that short amplicons replicate at the highest rate. The standard deviation of the copy number in panels **a** and **b** is 4%, while for panel **c**, it is approximated as $3.8\% + 1/\sqrt{copy}$, see Methods for details. Source data are provided as a Source Data file.

at higher template amounts over $6 \times 10^6$ copies, we notice that PCR is more prone to forming nonspecific products resulting in a smear, while SHARP produces clean and well-defined amplicons. We note that both HDA and tHDA are limited to only <150 bp amplicons[1,13,16].

SHARP kinetics (Fig. 2c) shows an exponential increase in the product amount. The gel image (Fig. 2d) shows one dominant band at 1463 bp corresponding to the desired product. For the template copy number below $6 \times 10^4$ and no template control (NTC), we observe a small amount of low molecular weight product that we attribute to primer dimers. The primer-dimers start accumulating between 20 and 25 min at 65 °C, as judged by the kinetics curve of NTC SHARP (Fig. 2c). The primer-dimer band is also observed in PCR but is not as pronounced at 30 cycles. We speculate that ATP hydrolysis by PcrA M6 helicase, which is stimulated by single-stranded DNA, may deplete ATP in 20–25 min at 65 °C, shifting the reaction balance toward primer-dimer formation when the initial template amount is small.

Quantitative or real-time PCR (qPCR) uses a metric called the cycle threshold (CT), the number of cycles after which the fluorescence of a PCR product can be detected above the background. Because we cannot directly determine the number of amplification cycles in SHARP, we instead determined the time it takes to detect the products in qSHARP, termed the detection time. The detection threshold was chosen so that the signal for non-template control (NTC) remains below the threshold for 25 min. Figure 2e displays the initial template copy number versus detection time. From the fit, assuming the exponential model, we determined the doubling time of $t_{1/2} \approx 1.5$ min for the 1464 bp amplicon. For a PCR with Phusion mix and 1.4 kbp amplicon, each cycle or doubling time is longer than SHARP. It

typically takes 2 to 3 min for a thermocycler to melt the duplex, bind primers, replicate, and change the temperature.

We also demonstrated SHARP with other template-primers sets, including ≈3 kbp amplicon (Fig. 3a), and ≈6 kbp amplicon (Fig. 3b). Fig. 3c shows a short 200 bp amplicon using λ-DNA as a template and the kinetics of the amplification reaction. For 200 bp amplicon we observe a gel band even at the single-digit template copy numbers per microliter. For each kinetics curve for 200 bp amplicon (Fig. 3c), we determined the detection time (Fig. 3d) and obtained the doubling time of $t_{1/2} = 0.83$ min or 50 s for the 200 bp product.

Figure 3e summarizes the SHARP detection limit for different amplicon lengths (200 bp, 1.4 kbp, 3 kbp, and 6 kbp). For short amplicons, under 200 bp, SHARP can detect and amplify only a few template molecules per microliter. As the DNA template length increases, SHARP needs more template molecules, Fig. 3e. This is also expected with a thermocycler-based PCR. Fig. 3f shows the doubling time ($t_{1/2}$) in SHARP for 3 different amplicon lengths. The short amplicon has the shortest doubling time of less than 1 min, while for longer amplicons, the time increases. For the 6 kbp amplicon, the low detection limit (Fig. 3b–e) resulted in an insufficient number of kinetics curves to determine the doubling time.

### Amplification of genomic DNA from human cells

We tested SHARP for targeted amplification of genomic DNA (gDNA). We purified gDNA from HEK293T cells and measured qSHARP amplification of a 474 bp DNA region near the *DMNT3B* gene in the presence of EvaGreen intercalating dye. The product started increasing above the detection threshold around 20 min at 60 °C. No template control (absence of gDNA) started showing non-specific amplification after

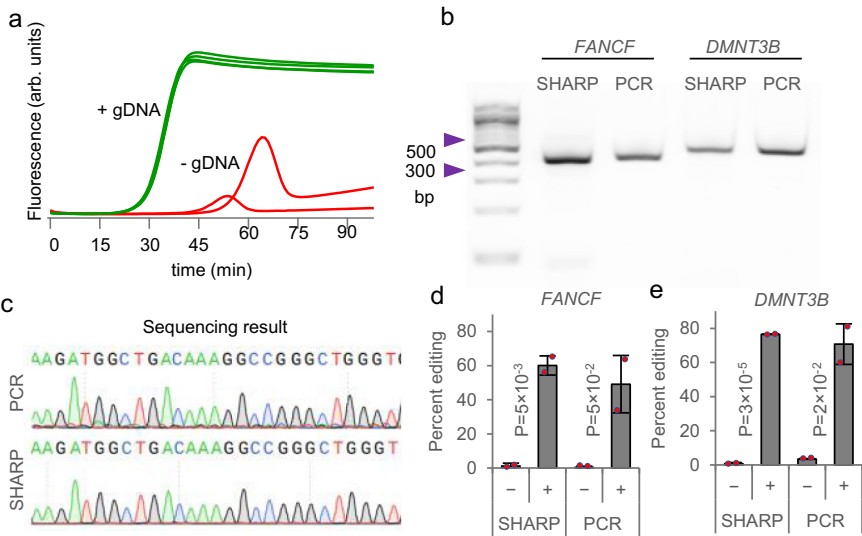

**Fig. 4 | SHARP applied to genomic DNA. a** The SHARP reaction containing gDNA template (+gDNA) or without template (−gDNA), along with primers intended to amplify a 474 bp region near the *DMNT3B* gene, was incubated at 60 °C. The extent of amplification was quantified by measuring DNA fluorescence using the EvaGreen dye as a function of time. **b** Agarose gel electrophoresis of genomic amplification product using SHARP versus PCR, amplifying a 410 bp region near *FANCF* or a 474 bp region near *DMNT3B*. **c** Sanger sequencing of genomic amplification product near *DMNT3B* using SHARP versus PCR. d-e Quantification of genome editing insertion-deletion mutations (indels) using SHARP versus PCR, near *FANCF* and *DMNT3B* as panel **b**. Signs '−' and '+' correspond to gDNA from cells without or with Cas9/gRNA, respectively. Cas9/gRNA has been introduced to two biologically independent cell cultures. Data are presented as mean values ± standard deviation. Based on *P*-values obtained with Student's *t*-test, SHARP quantified that the percent editing in cells with introduced Cas9/gRNA (+) is statistically significant relative to unedited (−) cells with the confidence level of 95%. Source data are provided as a Source Data file.

45 min, giving a clear separation in time between the specific product (+gDNA) and the delayed nonspecific product (-gDNA) (Fig. 4a). By quenching the reaction sooner, the nonspecific product can be avoided, like in other SHARP assays. In addition to the real-time detection, agarose gel electrophoresis and Sanger sequencing confirmed the correct SHARP product for both *DMNT3B* and another 410 bp region near *FANCF*. We also benchmarked SHARP against thermocycler-based PCR for the gDNA template, and they both led to the same product in the gel assays (Fig. 4b) and Sanger sequencing (Fig. 4c).

Apart from simple detection, we further determined that gDNA amplification using SHARP can quantify the level of CRISPR-Cas9 genome editing[19]. We delivered Cas9 in complex with guide RNA (gRNA) targeting either the *FANCF* or *DMNT3B* genes into HEK293T cells using electroporation[21], incubated for 72 h to allow the sufficient duration of Cas9 activity, and purified the gDNA from cells. SHARP amplification of the *FANCF* or *DMNT3B* regions containing the Cas9 cleavage site from the gDNA of edited cells, followed by Sanger sequencing and comparison to unedited cells, enabled characterization of insertion-deletion (indel) mutations induced by Cas9[22]. As expected, cells without Cas9/gRNA exposure exhibited background levels of indels, whereas cells with 72 h of Cas9/gRNA exposure exhibited greater than 50% indels, with results comparable to that of gold standard PCR (Fig. 4d, e). Together, these results demonstrate that SHARP can robustly amplify genomic DNA, with applicability as the PCR-free method for targeted quantification of genome editing outcomes at endogenous genomic sites[23].

**Other applications of SHARP**

SHARP is an isothermal alternative to PCR for many other common applications in the wet lab. We tested whether SHARP makes functional DNA that can be propagated in living cells. Starting from a 4012 bp template vector (Fig. 2a), we chose primers to include the vector backbone containing ampicillin resistance (see Methods), amplified the 3245 bp region with SHARP (Fig. 3a), and ligated the blunt ends. *E. coli* DH5α cells were transformed with this plasmid and

then plated. We grew 3 randomly picked colonies, extracted plasmid, linearized, and confirmed the correct 3245 bp product in all three (Fig. 5a). Therefore, *E. coli* can take SHARP-made plasmid, obtain ampicillin resistance, survive, and replicate. For applications in cloning and gene manipulation, the amplification fidelity plays an important role because it determines how accurate the DNAP replicates the desired template. The error rate of Bst-LF used in SHARP was estimated to be about $60 \times 10^{-6}$ errors/base[24], where 89–92 % of all errors are substitutions, 7–8% are deletion, and 1-3% insertions. Bst-LF is less prone to errors than Taq DNAP ($2 \times 10^{-4}$ to $2 \times 10^{-5}$ errors/base), but more prone to errors than Phusion[25].

We further tested whether SHARP could amplify sequences prone to secondary structure formation. We selected a 385-bp region containing 47 trinucleotide CAG repeats known to undergo multivalent intermolecular interactions in the single stranded form[26]. Trinucleotide CAG repeats are known to have low PCR yields[27,28]. In the example in Fig. 5b we show that a sequence containing $(CAG)_{47}$ does not inhibit SHARP reaction over a range of SSB concentrations.

**Temperature dependence**

Next, we tested the temperature dependence of the SHARP. With nCoV-2 sequence template and primers targeting the 155 bp region, we ran SHARP in the temperature range between 45 °C and 65 °C, and recorded the kinetics curves (Fig. 5c). The reaction is quenched after 30 min, and the product imaged on 2% agarose gel (Fig. 5d). For the reactions in the range 48.9 °C and 65 °C, we observed the 155 bp amplification product, while at lower temperatures amplification did not occur within 30 min. At 65 °C, it takes about 5 min to detect the product, while at 48.9 °C the amplification time increases to 25 min (Fig. 5c, inset). Therefore, SHARP with Bst-LF DNAP and specific primer works optimally at 65 °C, but it can also work at lower temperatures. Bovine Serum Albumin (BSA) and *E. coli* SSB used with SHARP at 65 °C are not extracted from thermophilic organisms, but they both support SHARP at elevated temperatures, suggesting they maintain their supporting roles. BSA is frequently used in PCR mixes and can withstand temperatures above 65 °C while early studies of *E. coli* SSB showed it

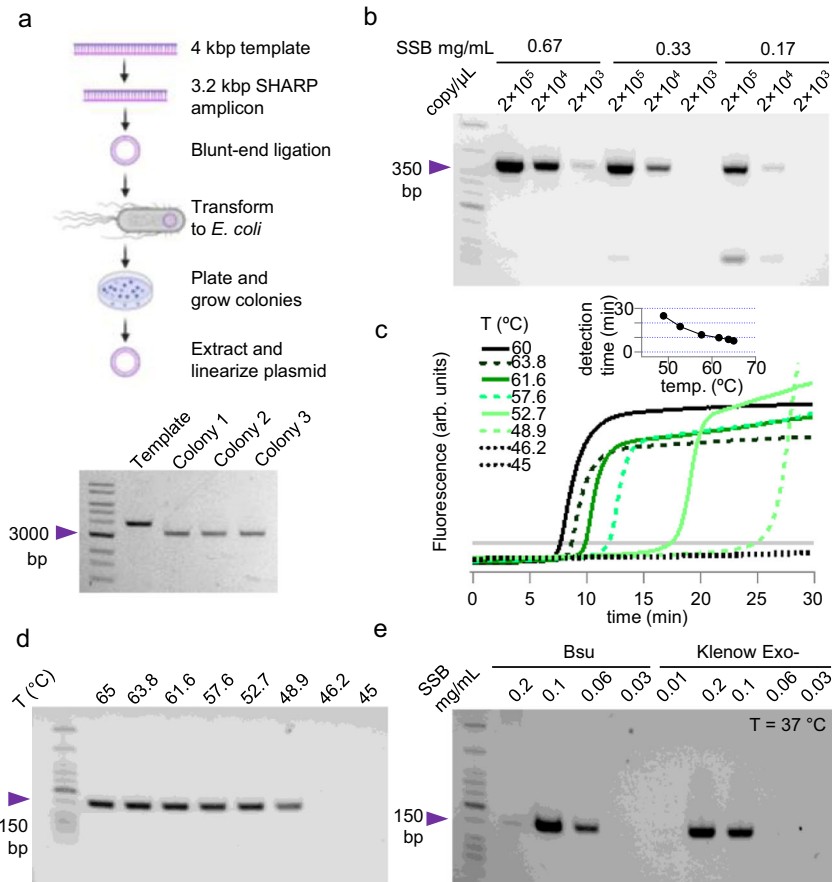

**Fig. 5 | Applications of SHARP. a** Transforming SHARP-made plasmid to *E. coli*. **b** SHARP amplifies sequence prone to forming secondary structure (CAG repeats) at different SSB concentrations. **c** SHARP kinetics curves at different temperatures, inset shows detection times. **d** SHARP product at different temperatures. **e** SHARP at 37 °C with Bsu and Klenow Exo⁻ DNAPs at variable SSB concentration. The standard deviation of the template copy number and SSB concentration is 4%, see Methods for details. Source data are provided as a Source Data file.

remains stable and active after boiling and exposure to high temperatures[29].

Bsu and Klenow Exo⁻ are better DNAP choices if SHARP must be carried out at 37 °C. Fig. 5e shows 155 bp product amplified at 37 °C with Bsu and Klenow Exo⁻ DNAPs over a range of SSB concentration. The reaction was quenched after 1 h without producing primer-dimers possibly because PcrA M6 ATPase activity is also lower at 37 °C than at 65 °C. We observed that Klenow Exo⁻ needs higher SSB concentration than Bsu for successful amplification. SHARP at 37 °C has two limitations, the amplicon length is less than 1 kbp and the amplification time can take more than 1 h even for highly concentrated template, but lower temperature may make SHARP useful for specialized applications.

Overall, we demonstrated with the examples in Figs. 2–4 that SHARP performs equally well or better than PCR according to the 8 criteria. (1) SHARP can generate up to 6000 bp amplicons in large quantities, unlike existing isothermal amplification methods generating only a few hundred base pair products. (2) SHARP amplification time is dependent on the template and primers, but it usually occurs within 5–30 min. (3) Primers optimized for PCR can be used with SHARP, unlike many other isothermal methods requiring more complex primers design. (4) SHARP detection limit is comparable to PCR, based on gel assays. (5) SHARP is also suitable for the real-time product monitoring in the presence of intercalating dye, as an isothermal alternative to qPCR. (6) The correct SHARP and PCR product appear as a single band on the gel; hence, it is straightforward to determine the product presence, length, amount, and sequence. Unlike SHARP and PCR, LAMP generates multi-band product, often containing more than

10 bands, making it challenging to interpret, quantify, and sequence the result. (7) SHARP product can be used in downstream applications in the same manner as PCR product. For example, it can be purified, sequenced, or used for cloning. (8) Even when the template is much longer than the amplicon in gDNA, SHARP is as specific as PCR and does not require initial heat denaturing steps.

**Engineering PcrA M6 helicase for SHARP**

The fast, processive, and thermostable PcrA M6 helicase plays an essential role in replacing the thermocycler and enabling SHARP. Here we use bulk, single-molecule, and in silico assays to explore the relation between the structure and activity of PcrA M5/M6 and other helicase candidates for SHARP. With a bulk FRET unwinding assay, we initially determined the unwinding activity of seven helicase candidates for SHARP[18]. Our FRET DNA construct (Fig. 6a) consists of 18 bp duplex with 3′ overhang labeled with the FRET pair of Cy3 and Cy5 in such a manner that FRET decreases upon DNA unwinding (Fig. 6a). A reference reaction using a highly-processive engineered Rep-X superhelicase[18] gave rapid FRET decrease upon adding ATP (Fig. 6b), recorded on Cary Eclipse fluorometer. Rep-X showed the highest unwinding activity at 37 °C, but it is not suitable for SHARP because its activity decreases at elevated temperatures, and it cannot unwind blunt-ended DNA. Rep-X showed no amplification with SHARP. We also tested *E. coli* UvrD helicase for SHARP because UvrD was used with HDA to amplify 100 to 200 bp fragments[13]. UvrD at 34 nM is nearly equivalent 10 nM Rep-X when it comes to unwinding the DNA construct (Fig. 6c) but the DNA underwent subsequent reannealing, shown as FRET recovery, likely due to ATP depletion. UvrD did not support

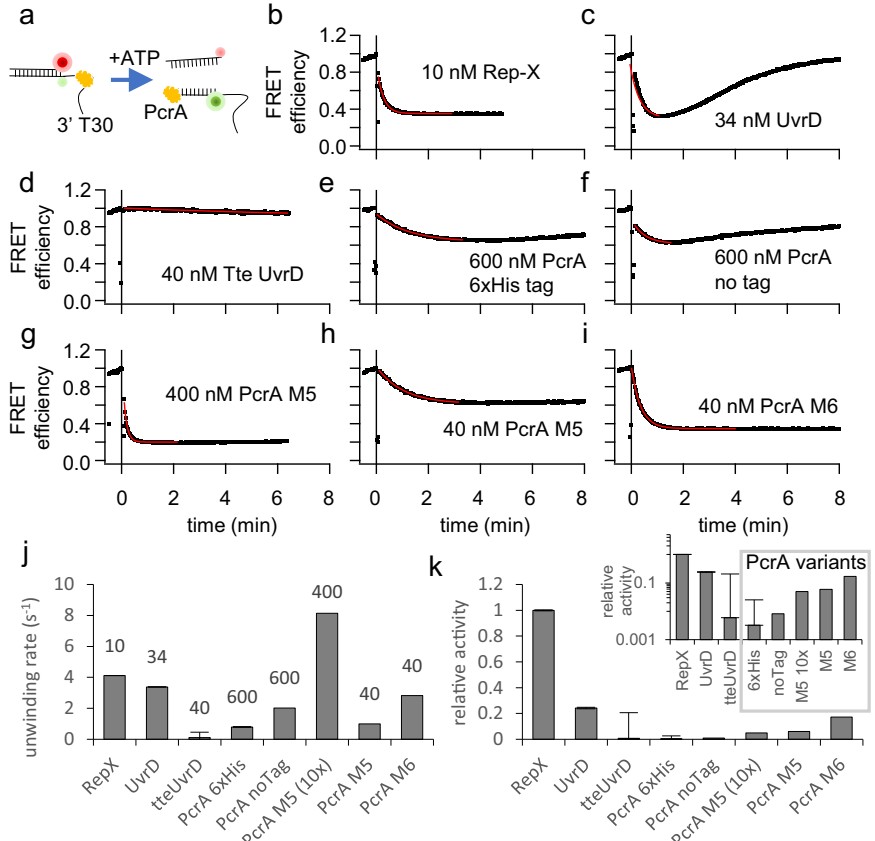

**Fig. 6 | Bulk unwinding activity of different DNA helicases at 37 °C.** Each helicase is at different concentration, while all other conditions are fixed. **a** Unwinding reaction of FRET-pair labelled DNA construct. **b** Rep-X superhelicase shows the highest unwinding activity even at a low concentration of 10 nM. **c** *E. coli* UvrD helicase at 34 nM shows high initial unwinding activity, but the reaction terminates quickly resulting in the DNA reannealing likely due to ATP depletion. **d** Thermostable Tte UvrD helicase at 40 nM shows very low unwinding activity at 37 °C. **e** Wild-type PcrA expressed with 6xHis tag shows low unwinding activity at 600 nM. **f** Wild-type PcrA without 6xHis tag shows low unwinding activity at

600 nM. **g** PcrA M5 mutant at 400 nM shows high unwinding activity. **h** PcrA M5 mutant at 40 nM. **i** PcrA M6 shows higher unwinding activity than M5 even at 40 nM concentration. SHARP uses nearly 40 nM of PcrA M6. **j** Unwinding rates obtained from the fit to FRET efficiency in **b** to **i**. **k** Unwinding activity relative to Rep-X is obtained by dividing unwinding rates in **j** by the concentration and normalizing. Inset shows relative activity on $\log_{10}$ axis. Data in **j** and **k** are presented as mean values ± standard error obtained from the fit. Source data and fitting parameters are provided as a Source Data file.

SHARP. Thermostable *Thermoanaerobacter tengcongensis* Tte UvrD[16] showed no unwinding activity at 37 °C (Fig. 6d) and no SHARP activity at 65 °C.

*Geobacillus stearothermophilus* PcrA helicase[30] showed low unwinding activity at elevated concentration of 600 nM (Fig. 6e, f) and it did not support SHARP. PcrA M5 mutant, introduced in[18], has two native cysteines mutated (C96A and C247A) (Fig. 7a) and two new cysteines introduced (N187C and L409C), and one additional mutation L384V. PcrA M5 was previously[18] constrained to its closed conformation with an external crosslinker such as bismaleimidoethane (BMOE) or 1,8-bismaleimido-diethyleneglycol (BM(PEG)₂). Here, we found that applying crosslinkers to M5 is not necessary. When PcrA is in its unwinding active, closed conformation, the distance between two residues N187 and L409 is ~4 Å[31]; hence, after mutating two residues to cysteines (N187C and L409C) a disulfide bond between C187 and C409 can form spontaneously and keep M5 in its active, closed conformation. In the Supplementary section, we further explored conditions for the disulfide bond formation in M5 and M6 mutants and tested how DTT affects SHARP. We found that an increased DTT concentration (>50 mM) inhibits SHARP, as expected because DTT breaks disulfide bonds essential for the superhelicase activity of our engineered enzyme in the absence of externally applied crosslinker (Supplementary Fig. 2a). DDT had minimal effect on SHARP activities of cysteine-crosslinked helicase with BM(PEG)₂ (Supplementary Fig. 2b). PcrA M5

at 400 nM showed higher unwinding activity (Fig. 6g) than PcrA, but the activity of M5 significantly decreased at 40 nM (Fig. 6h).

We introduced an additional H93A mutation to PcrA M5 to create M6 and further improve its unwinding speed and activity. An equivalent mutation in Rep helicase (H85A) makes Rep 2-3 times faster as a single-stranded DNA translocase (Fig. 2.9 in ref. [32]). In a bulk FRET assay (Fig. 6i), PcrA M6 shows up to 4 times higher unwinding activity than PcrA M5 at 40 nM. Both PcrA M5 and M6 support SHARP, but M6 is more active and consequently SHARP mix requires a lower concentration of M6 (30 to 90 nM) relative to M5 (over 100 nM). PcrA overexpression is toxic to *E. coli* cells, leading to low to moderate amounts of purified protein[30]; hence, we find M6 more practical than M5, because the same amount of purified protein enables more reactions. Fig. 6j summarizes the apparent unwinding rates of different helicases we tested. We also calculated normalized helicase activity by dividing the unwinding rate by the helicase concentration used (Fig. 6k inset) and plotted the normalized activity relative to Rep-X (Fig. 6k). Rep-X shows the highest unwinding activity, followed by UvrD, PcrA M6, PcrA M5, and PcrA in that order. Although PcrA M5/M6 are not as active as Rep-X or UvrD according to the FRET assay, only they supported SHARP.

Figure 7a illustrates locations of 6 point mutations in M6, mapped to the PcrA structure[31]. We further characterized the bulk unwinding activity of PcrA M6 on various DNA substrates using intercalating dyes

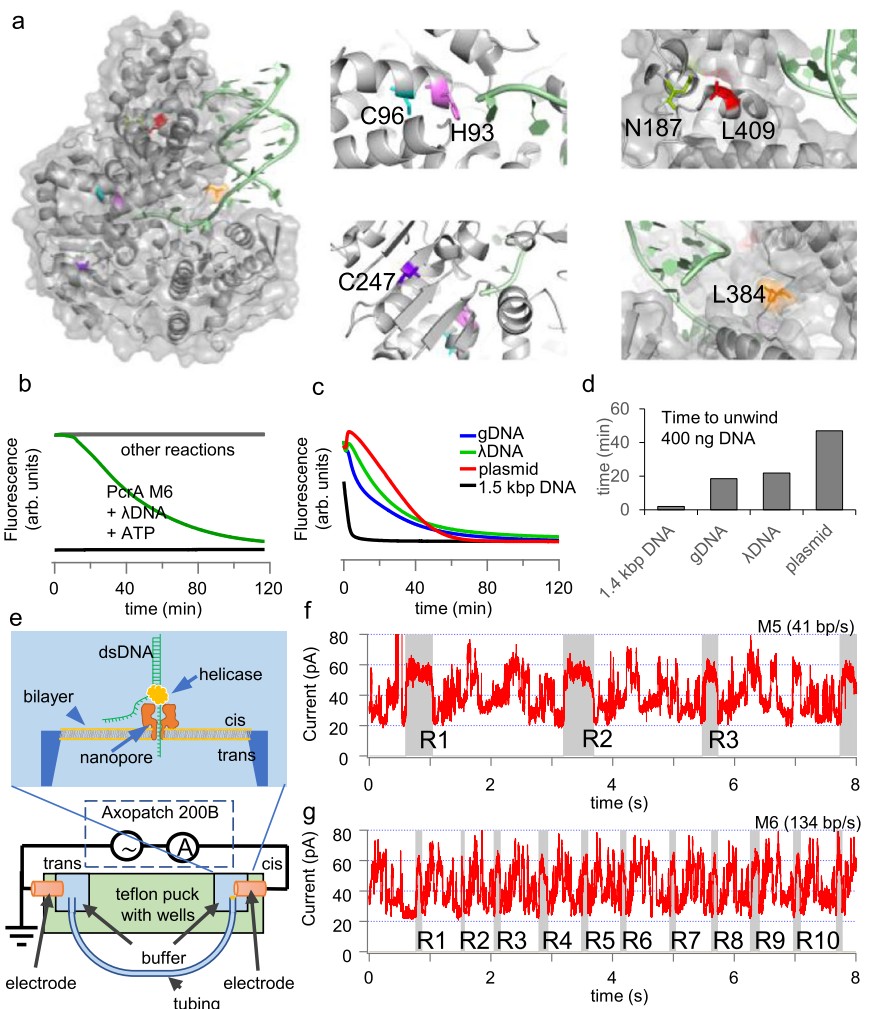

**Fig. 7 | Structure and characterization of PcrA M6 unwinding kinetics.**
**a** Structure of wild-type PcrA and location of point mutations H93A, C96A, N187C, C247A, L384V, L409C for improved activity. The PcrA structure was obtained from the UniPort database, accession number "P56255". **b** Unwinding of λDNA occurred in the presence of PcrA M6, SSB and ATP. Other reactions λDNA+SSB, PcrA M6+λDNA, PcrA M6 + ATP + λDNA showed no unwinding. **c** Unwinding of different DNA substrates concentrated at 20 ng/μL in the presence of PcrA M6, SSB, and ATP. **d** Time needed to unwind different DNA substrates in the presence of PcrA M6, SSB and ATP. **e** Scheme of the SPRNT setup where a helicase drives Spinach64 DNA through MspA nanopore. **f** and **g** Periodic electric current signals are used to determine the unwinding speed of PcrA mutants M5 and M6. The signal marked in gray and labeled by R1, R2, ...R10, shows the beginning of each repeat in the periodic spinach DNA sequence. By counting number of repeats in time, we determine the helicase speed. Supplementary Information contains the DNA sequence corresponding to each repeat. Source data are provided as a Source Data file.

at 37 °C (Fig. 7b). Unwinding is detected as a decrease in the fluorescence intensity over time. PcrA M6 at 40 nM with ATP did not unwind 48 kbp λDNA; however, in the presence of 0.2 mg/mL SSB, complete unwinding occurred (Fig. 7b). This ability to unwind long DNA is important for DNA amplification without an initial heat denaturing step, such as gDNA from human cells. PcrA M6 at 40 nM concentration could unwind 1.5 kbp linear DNA, long gDNA, λ DNA, and 4 kbp circular plasmid in the presence of 0.2 mg/mL SSB and 2 mM ATP (Fig. 7c). For the same mass concentration of DNA substrate used (20 ng/μL), PcrA M6 showed the fastest unwinding for 1.5 kbp linear DNA and showed unwinding activity for the circular plasmid albeit at the slowest rate among those tested, Fig. 7d.

For better understanding of the underlying mechanistic properties of PcrA M5/M6, we employed Single-molecule Picometer Resolution Nanopore Tweezers (SPRNT)[33] and molecular dynamic (MD) simulations. SPRNT assay shows our engineered superhelicase can unwind several thousands of base pairs after single binding to a DNA substrate and we find this property important for generating kilo-bp-long amplicons. SPRNT assay also directly measures the helicase unwinding speed and shows that the introduced mutation H93A

increases the helicase speed up to 4 times. This is consistent with the increased activity of PcrA M6 observed in the bulk FRET assay.

The SPRNT setup (Fig. 7e) consists of two wells separated by a phospholipid bilayer and connected with MspA nanopore inserted into the bilayer[33]. The well near the positive electrode contains the signal buffer (500 mM KCl and 50 mM HEPES pH 8.0), while the well near the negative electrode contains the reaction buffer (200 mM KCl, 50 mM HEPES pH 8.0, 5 mM MgCl2, and 2 mM ATP) and helicase-DNA construct[33] at room temperature (22 °C). The DNA construct has 64 repeats of spinach DNA sequence where each repeat is 103 bp longs[34]. We apply 180 mV electric potential difference between the electrodes and measure pA current with Axon Axopatch 200B instrument controlled by LabVIEW 2018. As the helicase unwinds the DNA and drives one strand into the nanopore, the electric current passing through the nanopore generates a repetitive signal, Fig. 7f. By counting the number of 103 bp-long repeats in time, we can determine helicase unwinding speed. For PcrA M5 in Fig. 7f, we observe 3.2 repeats during 8 s. For M6 we identified 10.4 repeats during 8 s. Figure S2c, d show PcrA M5 and M6 unwinding ≈1500 and ≈3800 nucleotides after single binding to DNA. By analyzing unwinding of ≈10^4 nucleotides, we find the average

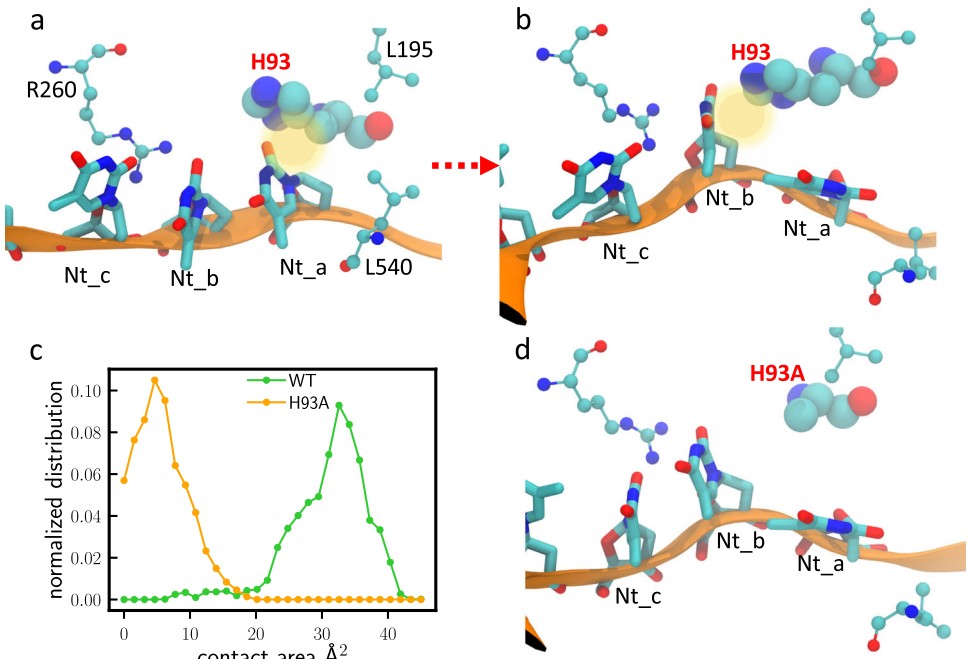

**Fig. 8 | MD simulations reveal that H93 plays an important role in DNA translocation. a** ssDNA-PcrA interactions at the initial (pre-translocation) state for WT PcrA. **b** ssDNA-PcrA interactions at the final (post-translocation) state for WT PcrA. **c** Distributions for the contact areas at the final state (green curve for Nt_b - H93; orange curve for Nt_b - H93A). **d** ssDNA-PcrA interactions at the final state for the H93A mutant.

speed of PcrA M5 and M6 of 40 bp/s and 134 bp/s, respectively, and conclude that at room temperature, mutation H93A increases the unwinding speed up to 4 times.

## MD simulations for the PcrA-DNA complex

We used MD simulations for the PcrA-DNA complex to further investigate what role H93 residue plays in PcrA translocation and unwinding. We observed that H93 served as a gating residue to control ssDNA translocation. As demonstrated in Fig. 8a, initially, H93 interacted with the outgoing nucleobase (Nt_a). Through the course of the simulation, H93 disengaged from Nt_a and eventually formed interaction with the next departing nucleobase (Nt_b) in the final state (Fig. 8b). Next, we carried out simulations with H93 mutated into alanine. H93A did not form close contact with Nt_b in the final state, showing a diminished role in gating (Fig. 8d). We measured the contact area between Nt_b and H93/H93A in the final state (Fig. 8c). The average contact area for Nt_b - H93 is 32 Å², much larger than that for Nt_b - H93A (5 Å²). Similarly in the initial state, the average contact area for Nt_a - H93 is 40 Å², much larger than the 25 Å² measured for Nt_a - H93A (Supplementary Fig. 3c). We thus suggest that the mutation H93A lowers the energy barrier and reduces the friction for DNA translocation, leading to a higher unwinding activity as also observed in both bulk unwinding and SPRNT assays.

Finally, we summarize what PcrA M6 properties are important for enabling SHARP. PcrA monomers are very good ssDNA translocases yet are ineffective as helicases in vitro[30]. With disulfide crosslinking, PcrA M5 and M6 become very effective helicase capable of unwinding several thousand base pairs after single binding to dsDNA. This property of the engineered helicase is important for generating kilobase-long amplicons in SHARP. PcrA plays an important role in plasmid replication and can unwind circular plasmids in the presence of the initiator protein RepD[35,36]; however, PcrA M5/M6 on its own requires a DNA substrate with 3′-ssDNA overhangs to initiate the unwinding[18,30], making it unsuitable for SHARP. We have shown that in the presence of SSB, PcrA M5/M6 can effectively unwind DNA with blunt ends and circular plasmids, and these abilities are likely to be essential for initiating each replication cycle in SHARP. PcrA M5 has reduced ATPase

activity relative to PcrA[18] and this property keeps ATP abundant even during prologued amplification times characteristic to reactions with low copy numbers of the initial template. The last important property to mention is the thermostability of PcrA M5/M6 and their ability to remain active in a very wide temperature range. This enables many future applications in molecular diagnostics, where adjusting for the annealing temperature of primers or probes improves the diagnostics specificity.

In this article, we engineered PcrA M6 helicase and introduced an isothermal amplification method named SHARP. SHARP eliminates thermal cycling from PCR, while keeping attractive PCR characteristics. SHARP is a robust and sensitive method in molecular diagnostics capable of detecting a small number of initial template molecules, while at the same time SHARP remains highly specific as it amplifies the correct product even in a long and diverse genomic DNA. SHARP can generate linear and several kilo-base pair long amplicons from a template and a set of two primers in under 30 min. Apart from replacing thermal cycling from PCR in molecular diagnostic and gene manipulation reactions such as cloning, SHARP can be used in determining the level of CRISPR-Cas9 genome editing inside human cells and it can also detect and amplify sequences prone to forming secondary structures. SHARP has the potential to become a universal tool for point-of-care detection methods in medicine, for simplifying many procedures in research labs, and for enabling applications such as environmental screening, farming pest detection, and other future consumer-focused diagnostic home tests.

## Methods

### SHARP reaction mix

This section contains information about SHARP reaction components and their concentrations. We overexpress and purify 3 principal enzymes, Bst-LF DNAP, *E.coli* SSB, and PcrA M6 helicase, while other components are purchased. We use following enzymes for SHARP with stock concentrations: SSB (9 mg/mL), PcrA M6 or M5 helicase (0.2 mg/mL), Bst-LF DNAP (1.5 mg/mL), PPase (2000 units/mL, NEB catalog # M0296S). Other stock components are dNTP's (10 mM each), ATP (100 mM), Evagreen dye (20X from Biotium # 31000), DTT

(Dithiothreitol, 100 mM in water). The 10X reaction buffer contains 500 mM Potassium acetate, 200 mM Tris-acetate, 100 mM Magnesium acetate, 1 mg/mL BSA, pH 7.9. Primer stock concentrations are at 10 or 20 $\mu$M, while the template concentration is variable. Table 1 contains volumes and stock concentrations of our purified enzymes. For purified enzymes, batch-to-batch variation in activity is common, so we suggest titrating enzyme volumes around values in Table 1 and monitoring qSHARP activity to find the optimal enzyme ratio for the specific batch. The optimal ratio is the one giving fastest SHARP kinetics.

For each SHARP reaction, we prepare 40 $\mu$L total volume and split the volume to two separate wells on a 96-well plate to have two independent fluorescence readouts of the same reaction. We monitor SYBR (EvaGreen) intensity in real-time every 10 s for each well on BioRad CFX96 machine run by CFX Manager software version 3.1. For all reactions except the reaction with gDNA, we separately prepare Component 1 and Component 2 and mix them together on the cold (4 °C) plate, before the temperature is increased to 65 °C and the recording of fluorescence starts. After incubation, we purify the product with Qiagen PCR cleanup kit and test the product on the 0.8% or 2% agarose gel.

## Primers and templates

Primers are synthetized by Integrated DNA technologies (IDT). DNA template vector containing 2019 coronavirus nCoV-2 N protein sequence (2019-nCoV_N_Positive Control, Catalog #10006625) is also purchaised from IDT. For PCR and SHARP product of 1.4 kbp in Fig. 2 we use and nCoV-2 N template vector and M13 primers:

5'-CCCAGTCACGACGTTGTAAAACG (forward) and

5'- AGCGGATAACAATTTCACACAGG (reverse)

For 3.2 kbp product in Fig. 3a we use and nCoV-2 N template and primers:

5'- AATTTTGGGGACCAGGAAC (forward) and

5'- TCTGGTTACTGCCAGTTGAATCTG (reverse) These primers also include the vector backbone containing ampicillin resistance.

The $\lambda$-DNA amplicon in Fig. 3c uses $\lambda$-DNA from NEB (catalog # N3011S) as a template and following primers generating 200 bp amplicon:

5'-CGGCTTCTGACTCTCTTTCC (forward) and

5'- TTCCTTCAAGCTTTGCCACA (reverse).

For testing SHARP with CAG repetitive sequence in Fig. 5b, we use pBluescript-CTG-47 sequence from[26] deposited on Addgene #99150 and M13 primers given above.

For testing the temperature dependence of SHARP in Fig. 5d, e, we use nCoV-2 N template and following primers generating 155bp product:

5'- AATTTTGGGGACCAGGAAC (forward) and

5'- GCACCTGTGTAGGTCAAC (reverse).

Primers used with gDNA from human cells are

5'- AGTTCGCTAATCCCGGAACT (FANCF_F)

5'- AGTTGCCCAGAGTCAAGGAA (FANCF_R)

5'- CCAGTGGTTCAATGGTCATCC (DNMT3B_F)

5'- GGCCAGTGAAATCACCCTG (DNMT3B_R) and

RNA sequences used for CRISPR-Cas9 editing are

tracrRNA 5'- AGC AUA GCA AGU UAA AAU AAG GCU AGU CCG UUA UCA ACU UGA AAA AGU GGC ACC GAG UCG GUG CUU U

crRNA_DNMT3B 5'- GGC ACU GCG GCU GGA GGU GGG UUU UAG AGC UAU GCU

crRNA_FANCF 5'- GCU GCA GAA GGG AUU CCA UGG UUU UAG AGC UAU GCU

## DNA unwinding assays

The FRET unwinding assay uses the following buffer: TRIS-HCl pH 8.0, 10 mM MgCl2, 50 mM NaCl, 1 mM ATP, 1% BSA. DNA concentration is at 5 nM and the helicase concentration varies. The reaction is performed in 0.2 mL cuvette. The fluorescence spectrophotometer (Varian model Cary Eclipse) excites the sample at 550 nm and records the fluorescence at 570 nm (green, Cy3) and 667 nm (red, Cy5).

Unwinding of longer DNA substrates in the presence of EvaGreen intercalating dye is carried out in 20 $\mu$L volume on Biorad CFX96 qPCR machine in the same buffer as the SHARP reaction (50 mM Potassium acetate, 20 mM Tris-acetate, 10 mM Magnesium acetate, 0.1 mg/mL BSA, pH 7.9). The reaction mix is prepared on ice, and the qPCR machine is precooled to 4 °C. Upon placing the 96-well plate on the qPCR machine, the machine starts recording and then rapidly increases the temperature to 37 °C to activate the enzymes and start the reaction.

## Protein overexpression and purification

PcrA M6 helicase with 6xHis tag: PcrA M6 helicase, PcrA, PcrA M5 and all mutants are purified as described previously[18,37] using standard Ni-NTA purification column, followed by single-stranded DNA cellulose column. Briefly, we use pET-11b vector containing PcrA M6 sequence between NdeI and BamHI sites. The vector including 6x-His tag on N terminus has been synthetized by GenScript and GenScript synthetized all other point mutations. We transform the vector to *E. coli* BL21(DE3) pLysS. Cells are grown at 37 °C in the presence of ampicillin and chloramphenicol, moved to 18 °C when OD600 reaches 0.3 and induced at OD600 = 0.5 with 0.5 mM IPTG and harvested after an overnight incubation at 18 °C. Cell pellets, previously stored at −80 °C were resuspended in the lysis buffer (50 mM Tris, 5 mM Imidazole, 200 mM NaCl, 20% Sucrose, 15% Glycerol, 0.5 mg/mL Lysozyme, pH 7.6) and sonicated followed by centrifugation at 35,000 g. The Ni-NTA agarose resin is preequilibrated with the wash buffer (50 mM Tris, 5 mM Imidazole, 150 mM NaCl, 25% (v/v) Glycerol, pH 7.6). 40 mL of cell lysate supernatant is added to 2 mL of equilibrated Ni-NTA resin and incubated for 1 h at 4 °C with constant stir mixing by inverting the 50 ml tube. After 1 h, the resin is gently centrifuged at 1000 g for 2 min, supernatant is discarded carefully, and the tube with the protein-loaded resin is refilled with the cold (4 °C) wash buffer. The batch wash is repeated 3 times, protein-loaded resin is then poured into a disposable gravity flow column, washed with 20 mL of cold wash buffer, and eluted with elution buffer made by dissolving 200 mM imidazole in wash buffer. The protein is then loaded to single-stranded DNA cellulose column, washed with buffer (100 mM NaCl, 50 mM Tris, 1 mM EDTA, 20% (v/v) Glycerol), and eluted with (1 M NaCl, 50 mM Tris, 1 mM EDTA, 20% (v/v) Glycerol). The presence of the 6xHis does not affect any downstream application. The protein concentration was always kept below 4 mg/ml (≈50 mM) to avoid aggregation, and the final PcrA M6 protein was stored at −80 °C or −20 °C in the storage buffer containing 600 mM NaCl, 50 mM TRIS pH 7.6, and 50% glycerol. This protocol leads to a diluted PcrA M6 to between 0.2 and 1 mg/mL, if necessary, one can concentrate protein with membrane filtering.

PcrA helicase without 6xHis tag: Plasmid expressing wild-type PcrA helicase without any tag is a gift from Tim Lohman lab. Obtaining the cell lysate is done in the same way as in the previous protocol. The cell lysate is also spun down, but supernatant is mixed with 0.7 volume of saturated ammonium sulfate solution. Ammonium sulfate precipitates PcrA, and precipitated PcrA is collected by spinning at 5000 g. We resuspended PcrA in the wash buffer (50 mM Tris, 5 mM Imidazole, 150 mM NaCl, 25% (v/v) Glycerol, pH 7.6), spun it down, and loaded the supernatant to ssDNA-cellulose column. The rest of the protocol is the same as for PcrA with the 6xHis tag.

Single Stranded binding protein (SSB): *E. coli* SSB is purified without tag through polymin-P and ammonium sulfate precipitation, followed by the heparin Sepharose column, like the protocol by Lohman et al.[38]. SSB sequence is cloned into pET21a vector using the NdeI and BamHI sites and we obtained the original vector from Tim Lohman. We transform the vector to BL21(DE3) cells, pick collonies, grow at 37 °C, induce overexpression with 0.5 mM IPTG at OD600 = 0.5 and grow for additional 5 h and collect the pellet. SSB expression levels are

usually very high, often SSB makes over 60% of the total cell protein. The cell pellet is resuspended in the lysis buffer (50 mM Tris, 200 mM NaCl, 20% Sucrose, 15% Glycerol, 0.5 mg/mL Lysozyme, pH 7.6) and sonicated, followed by the centrifugation at 35,000 g for 30 min. The supernatant containing soluble SSB is collected, and Polymin-P of the final concentration 0.2% is added to the supernatant to precipitate SSB. Precipitated SSB is collected by spinning at 4000 $g$ and then resuspended in buffer containing 50 mM TRIS pH 8.3, 20% glycerol, 1 mM EDTA, 400 mM NaCl, and stir mixed at 4 °C for 30 min. To remove undissolved protein, we spin at 10,000 g for 20 min and collect the supernatant containing soluble SSB. Finally, we add solid ammonium sulfate to give the final concentration of 150 g/L, precipitate SSB, and collect the pellet containing SSB after centrifugation at 12,000 for 30 min. The pellet is resuspended in 50 mM TRIS pH 8.3, 20% glycerol, 1 mM EDTA, 200 mM NaCl, stir mixed at 4 °C for 30 min, centrifuged at 18,000 $g$ for 20 min, and filtered through 200 $\mu$m filter. We equilibrate heparin-Sepharose column with a wash buffer (50 mM TRIS pH 8.3, 20% glycerol, 1 mM EDTA), dilute the SSB solution in the wash buffer five times and load on the column. Next, the bound SSB is washed with 50 to 100 mL wash buffer and eluted with NaCl gradient from 100 mM to 1 M. The final SSB solution is dialyzed against the storage buffer (20 mM Tris, pH 8.1, 50% Glycerol, 0.5 M NaCl, 1 mM EDTA, 1 mM BME). We obtain SSB concentrated between 6 and 9 mg/mL. The SHARP reaction uses highly concentrated SSB; hence, it is important to use SSB stock concentrated above 5 mg/mL.

Bst-LF DNA polymerase: Bst-LF purification protocol uses Ni-NTA resin. We use Bst-LF DNAP expressing vector from Andrew Ellington's lab available through addgene.org/145799/ and follow a modified version of the protocol described in[39] for the overexpression and purification. We transform Bst-LF expressing plasmid containing 6xHis tag on N terminus to expressing cells, select individual collonies, grow medium until $OD$600 reaches 0.5 and induce overexpression with 200 ng/ml anhydrotetracycline (aTC) or 100 ng/ml of tetracycline. The total culture volume of 500 mL is grown at 25 degrees for 12 h and spun down to form the pellet kept at -80 degrees. The pellet is resuspended in the Lysis buffer (20 mM Tris pH 7.4, 300 mM NaCl, 0.1% Tween-20, 10 mM imidazole, 1x EDTA-free protease inhibitor tablet, 0.5 mg/mL Lysozyme) in the cold room and sonicated to lyse the cells. Lysed cells are centrifuged at 35,000 g for 30 min at 4 °C and the supernatant is collected. We heat treat the lysate at 65 °C for 10 min, cool down on ice for 10 min at 4 °C, and filter using 60 mL syringe with 200 $\mu$m filter. The Ni-NTA agarose resin is preequilibrated with the wash buffer (20 mM Tris, pH 7.4, 300 mM NaCl, 0.1% Tween-20, 40 mM imidazole). 40 mL of filtered lysate is added to 2 mL of equilibrated Ni-NTA resin and incubated for 1 h at 4 °C with constant stir mixing by inverting the 50 ml tube. After 1 h, the resin is gently centrifuged at 1000 g for 2 min, supernatant is discarded carefully, and the tube with the resin refilled with the wash buffer. The batch wash is repeated 3 times, protein-loaded resin is then poured into a disposable gravity flow column, washed with 10 mL of wash buffer and eluted with a buffer containing 20 mM Tris pH 7.4, 300 mM NaCl, 0.1% Tween-20, 250 mM imidazole in 500 $\mu$L fractions. Collected fractions are tested on the gel, and dialyzed against the storage buffer (50% Glycerol, 10 mM Tris pH 7.4, 100 mM KCl, 1 mM DTT, 0.1 mM EDTA, 0.5% Tween-20, 0.5% Triton-X100) overnight. Small aliquots of Bst-LF DNAP concentrated at 1.5 mg/mL are kept at −80 °C for long term storage or at −20 °C for daily use. Optimal Bst-LF concentration in the SHARP reaction is between 0.0015 and 0.0075 mg/mL.

SpCas9 purification: SpCas9 purification was conducted as previously reported[21]. BL21-CodonPlus (DE3)-RIL competent cells (Agilent Technologies 230245) were transformed with Cas9 plasmid (Addgene #67881) and inoculated in 25 mL of LB-ampicillin media. The bacteria culture was first allowed to grow overnight (37 °C, 220 rpm) and then transferred to 2 L of LB supplemented with ampicillin and 0.1% glucose until OD600 of ~0.5. Subsequently, the cells were induced with IPTG at a final concentration of 0.2 mM and maintained overnight at 18 °C. The bacteria cells were pelleted at 4500 × $g$, 4 °C for 15 min and resuspended in 40 mL of lysis buffer containing 20 mM Tris pH 8.0, 250 mM KCl, 20 mM imidazole, 10% glycerol, 1 mM TCEP, 1 mM PMSF, and cOmplete™ EDTA-free protease inhibitor tablet (Sigma-Aldrich 11836170001). This cell suspension was lysed using a microfluidizer and the supernatant containing Cas9 protein was clarified by spinning down cell debris at 16,000 × $g$, 4 °C for 40 min and filtering with 0.2 $\mu$m syringe filters (Thermo Scientific™ F25006). Four millimeters Ni-NTA agarose bead slurry (Qiagen 30210) was pre-equilibrated with lysis buffer. The clarified supernatant was then loaded at 4 °C. The protein-bound Ni-NTA beads were washed with 40 mL wash buffer containing 20 mM Tris pH 8.0, 800 mM KCl, 20 mM imidazole, 10% glycerol, and 1 mM TCEP. Gradient elution was performed with buffer containing 20 mM HEPES pH 8.0, 500 mM KCl, 10% glycerol, and varying concentrations of imidazole (100, 150, 200, and 250 mM) at 7 mL collection volume per fraction. The eluted fractions were tested on an SDS-PAGE gel and imaged by Coomassie blue (Bio-Rad 1610400) staining on GE Amersham Imager 600. To remove any DNA contamination, 5 mL HiTrap Q HP (Cytiva 17115401) was charged with 1 M KCl and then equilibrated with elution buffer containing 250 mM imidazole. The purified protein solution was then passed over the Q column at 4 °C. The flow-through was collected and dialyzed in a 10 kDa SnakeSkin™ dialysis tubing (Thermo Fisher 68100) against 1 L of dialysis buffer (20 mM HEPES pH 7.5, and 500 mM KCl, 20% glycerol) at 4 °C, overnight. Next day, the protein was dialyzed for an additional 3 h in fresh 1 L of dialysis buffer. The final Cas9 protein was concentrated to 10 $\mu$g/$\mu$L using Amicon Ultra-15 Centrifugal Filter Unit, Ultracel-10 (Millipore Sigma UFC901008), aliquoted, and flash-frozen and stored at −80 °C.

## Delivery of SpCas9 targeting human genomic DNA via electroporation

SpCas9 was delivered to cell as previously reported[21]. 1.2 $\mu$L of 100 $\mu$M crRNA was mixed with 1.2 $\mu$L of 100 $\mu$M tracrRNA (Integrated DNA Technologies) and heated to 95 °C for 5 min in a thermocycler, then allowed to cool on benchtop for 5 min. To form the RNP complex, 1.7 $\mu$L of 10 $\mu$g/$\mu$L of purified Cas9 or AncBE4max was mixed with the annealed 2.4 $\mu$L 50 $\mu$M cr:tracrRNA, 0.9 $\mu$L of 1x PBS was mixed in, then the total 5 $\mu$L solution was incubated for an additional 20 min at room temperature to allow for RNP formation.

HEK293T cells were purchased from ATCC, (catalog # CRL-3216) and properly maintained to a confluency of ~90% prior to electroporation. For commercial HEK293T cell line, we did not perform authentication. 800,000 cells were trypsinized with 5 min incubation in the incubator, then 1:1 of DMEM complete was added to inactivate trypsin. This mixture was centrifuged (3 min, 200 × $g$), supernatant removed, followed by resuspension of the cell pellet in 1 mL PBS, centrifugation (3 min, 200 × $g$), and finally complete removal of supernatant. 20 $\mu$L of nucleofection solution (3.6 $\mu$L of Supplement solution mixed with 16.5 $\mu$L of SF solution from SF Cell Line 4D-Nucleofector™ X Kit S) (Lonza) was mixed thoroughly with the cell pellet. The 5 $\mu$L RNP solution was mixed in along with 1 $\mu$L of Cas9 Electroporation Enhancer (Integrated DNA Technologies). The entirety of the final solution was transferred to one well of a provided 16-strip cuvette. Electroporation was then performed according to the manufacturer's instructions on the 4D-Nucleofector™ Core Unit (Lonza) using code CA-189. Some white residues may appear in the cell mixture after electroporation, but that is completely normal. DMEM complete was added before plating to culture wells pre-coated with 1:100 collagen. 100k cells were plated to a 12-well plate and incubated for 3 days (72 h) before genomic DNA extraction.

## Harvesting genomic DNA from HEK293T cells

HEK293T cells were rinsed off 12-well plates using PBS, spun down at 200 × $g$ for 3 min, resuspended in 200 $\mu$L PBS. Genomic DNA was

**Table 2 | Premixed SHARP components for amplification of genomic DNA**

| RxnMix (2X): | $\mu$L | EnzMix (2X): | $\mu$L |
| --- | --- | --- | --- |
| 10X Buffer | 4 | SSB (9 mg mL⁻¹) | 0.8 |
| dNTPs (10 mM each) | 2 | Bst-LF (1.5 mg mL⁻¹) | 0.2 |
| ATP (100 mM) | 2 | PcrA M6 (0.2 mg mL⁻¹) | 2 |
| DTT (100 mM) | 4 | Nuclease free water | 1 |
| Total | 12 | Total | 4 |

isolated from the cell suspension using DNeasy Blood & Tissue Kit (Qiagen 69506) according to the manufacturer's instructions, with elution into 50-100 $\mu$L of the supplied AE buffer.

## SHARP amplification of genomic DNA

To form the 20 $\mu$L SHARP reaction mix for amplifying the *DNMT3B* region, mix 9 $\mu$L water, 1 $\mu$L primer mixture, 1 $\mu$L genomic DNA, 6 $\mu$L RxnMix, 1 $\mu$L EvaGreen, and 2 $\mu$L EnzMix (see Table 2. for RxnMix and EnzMix components). To form the 20 $\mu$L SHARP reaction mix for amplifying the *FANCF* region, double the primer mixture to 2 $\mu$L, and halve the EnzMix to 1 $\mu$L. For measuring change in DNA amplification over time, incubate at 60 °C constant temperature for 2 h in a CFX Connect (Bio-Rad), taking fluorescence measurements every minute. Reaction cleanup was performed using QIAQuick PCR Purification Kit (Qiagen 28104) following the manufacturer's instructions with elution into 40 $\mu$L of the supplied EB buffer.

## PCR amplification of genomic DNA

Genomic DNA samples were amplified with PCR using Q5 Hot Start High-Fidelity 2X Master Mix (New England BioLabs M0494) in 10 $\mu$L reactions (3 $\mu$L water, 1 $\mu$L 10 uM primer mixture, 1 $\mu$L genomic DNA, 5 $\mu$L 2x master mix). Primer pairs for all sequences are listed in Table S1. For example, the primer set for amplifying around the *DNMT3B* target site is DNMT3B_F and DMNT3B_R. Thermocycler settings were 98 °C for 30 s, then 35 cycles of (98 °C 10 s, 67 °C 10 s, 72 °C 20 s), then 72 °C 2 min, then 4 °C hold. After PCR, cleanup was performed using QIAQuick PCR Purification Kit (Qiagen 28104) following the manufacturer's instructions with elution into 40 $\mu$L of the supplied EB buffer.

## Sanger sequencing

Ten microliters of the purified amplification product (from PCR or SHARP) was mixed with 5 $\mu$L of 5 nM forward amplification primer (DNMT3B_F, for example), and submitted to Genewiz for Sanger sequencing. Insertion-deletion mutations were calculated using TIDE analysis at https://tide.deskgen.com/.

## Molecular dynamic (MD) simulation

The DNA-bound PcrA complex was built based on the crystal structure "3PJR". The 3' poly(dT) ssDNA tail was extended by 10 nucleotides. The system was solvated in a water box with 150 mM NaCl. All simulations were performed using NAMD 2.13[40] with the CHARMM36 force field[41]. Full-system periodic electrostatics was calculated using the Particle Mesh Ewald method with a 12-Å cutoff applied to non-bonded interactions. The system was minimized with the conjugate gradient algorithm and then heated from 0 to 310 K in 100 ps while applying 50 kcal/(mol Å²) harmonic position constraints to the non-hydrogen atoms. A subsequent 10-ns simulation in the isothermal-isobaric ensemble was performed at 1 bar and 310 K while keeping the harmonic constraints at 1 kcal/(mol Å²). The Nose-Hoover method was used for pressure control, and Langevin dynamics was applied to maintain the temperature. The SHAKE algorithm was used to constrain bonds involving hydrogen atoms. The system was further equilibrated for 15 ns in the canonical ensemble at 310 K, with the harmonic constraints gradually reduced to

zero. Finally, a production run in the canonical ensemble was conducted for 75 ns until the system reached a stable final state.

Analysis and visualization of the simulation results were carried out with VMD[42]. To evaluate the contributions of individual PcrA residues during translocation, we calculated the contact area $\sigma$ between PcrA residues and nucleobases via $\sigma = (S_{prot} + S_{na} - S_{prot+na})/2$. Here, $S_{prot}$, $S_{na}$, and $S_{prot+na}$ denote the respective solvent-accessible surface areas of protein, DNA, and protein-DNA complex. The first and last 10 ns of the production run were used to calculate the contact-area distributions for the initial and final states, respectively (green curves in Fig. 8c and Supplementary Fig. 3c). To study the effects of the H93A mutant, histidine 93 was mutated into alanine in the initial and final configurations of the WT trajectory. Starting from these two configurations, MD simulations under the same conditions as WT were performed for 10 ns each. Similarly, the distributions for the contact areas between H93A and ssDNA nucleobases (Nt_a and Nt_b) were calculated for the two H93A trajectories (orange curves in Fig. 8c and Supplementary Fig. 3c). Kernel density estimate was used to obtain the normalized distributions for the contact areas.

## Statistics & reproducibility

The DNA template copy number (copy/$\mu$L) is determined in the following procedure. With Thermofisher NanoDrop 2000 instrument, we measure the concentration of DNA template stock in ng/$\mu$L, calculate the molar concentration in mol/L, and further express 1 mol as $N_A$ DNA copies, where $N_A$ is Avogadro number. This procedure gives DNA template copy number per $\mu$L. We make serial dilutions of stock. The accuracy of NanoDrop instrument for determining the concentration is 3%, while the pipetting accuracy is 0.8%. The uncertainty in estimating the DNA template copy number, we give as the standard deviation $\sigma_{copy}$. We estimate the uncertainty as $\sigma_{copy} = 3.8\% copy + \sqrt{copy}$ or relative uncertainty $\sigma_{copy}/copy = 3.8\% + 1/\sqrt{copy}$. This uncertainty has instrumental and statistical contribution. The instrumental contribution is affected by the 3% accuracy of NanoDrop 2000 used to measure the DNA template concentration and 0.8% pipetting accuracy. The statistical contribution $(1/\sqrt{copy})$ is negligible for high template copy numbers, while for single digit template copy numbers, ie assay with $\lambda$DNA, it dominated the result.

We used Microsoft Excel function for two-tailed homoscedastic Student's $t$-test to determine whether percent editing in cells with introduced Cas9/gRNA (+) is statistically significant relative to unedited (-) cells with a confidence interval of 95%. For both *FANCF* and *DMNT3B* regions amplified by SHARP, obtained $P$ values are $P = 5 \times 10^{-3}$ and $P = 3 \times 10^{-5}$, suggesting that the percent editing in cells with introduced Cas9/gRNA is statistically significant. For *FANCF* region amplified by PCR, we obtained $P = 5 \times 10^{-2}$, and concluded the quantified percent editing is not statistically significant, while for *DMNT3B* region amplified by PCR, we obtained $P = 2 \times 10^{-2}$ suggesting significant editing. The calculation of $P$ values is provided in the Source Data Files.

No statistical method was used to predetermine the sample size. No data were excluded from the analyses. The experiments were not randomized. The Investigators were not blinded to allocation during experiments and outcome assessment.

## Reporting summary

Further information on research design is available in the Nature Research Reporting Summary linked to this article.

## Data availability

Source data are provided with this manuscript. Gel images, qSHARP, fluorescence, nanopore, and sequencing data generated in this study are provided in the Source Data file. Plasmid and protein sequence data for the engineered helicase are provided in the Supplementary Information. Source data are provided with this paper.

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

## Acknowledgements

SSB and PcrA plasmid were a gift from Tim Lohman. Bst-LF expressing plasmid was a gift from Andrew Ellington (Addgene plasmid # 145799). pBluescript-CTG-47 was a gift from Ron Vale (Addgene plasmid # 99150). pHO4d-Cas9 was a gift from Michael Nonet (Addgene plasmid # 67881). We thank Elizabeth Weiland and Alexander Kozlov in the Lohman

laboratory for the advice on SSB overexpression and purification. This work was supported by grants from the JHU COVID-19 Research Response Program, National Institutes of Health (R35 GM 122569 to T.H.), and the National Science Foundation (MCB 2031094 to T.H.). M.G. received NSERC Canada postdoctoral fellowship. T.H. is an investigator with the Howard Hughes Medical Institute.

## Author contributions
M.G. performed experiments and analysis; T.H. and M.G. wrote the draft; R.Z. performed gDNA experiments; W.M. did MD simulations; J.K tested the effect of DTT; J.Y. and Soni. M. did preliminary tests; CY.L. and Sua.M. optimized buffer condition; TW.L. carried single-molecule controls

## Competing interests
Johns Hopkins Technology Venture Office have submitted a patent application to the United States patent office for inventors T.H. and M.G. pertaining to the bioengineered enzyme and isothermal amplification method aspects of this work (application number 63/304,189 filed on Jan 28. 2022). The remaining authors declare no competing interests.
