## [Peer Review File · Nature Communications]

REVIEWER COMMENTS

Reviewer #1 (Remarks to the Author):

This manuscript describes the application of an engineered superhelicase to develop an isothermal DNA amplification method, which authors named SHARP. This amplification method is similar to a commercially available one, Helicase Dependent Amplification (HDA), as the authors acknowledge. The main novelty is the use of a different helicase, a mutated one with additional modifications compared to a Superhelicase already described by the authors in reference 17. The Superhelicase is characterized by the ability to progressively unwind thousands of base pairs of DNA, with the final result of amplifying DNA sequences up to 6000 bp. HDA, which uses another helicase with limited unwinding processivity, is recommended for the amplification of short sequences (<150 nt). Within this framework, this is a significant contribution, demonstrating the higher processivity of the engineered helicase to develop an isothermal amplification DNA assay with performance comparable to PCR but at a constant temperature (optimal 65 °C). The authors demonstrated multiple applications for SHARP, including the amplification of linear DNA templates up to 6000 bp, the amplification of genomic DNA (474 amplicons), the quantification of CRISPR-Cas9 genome editing, the production of functional DNA that can be propagated in *E. coli* and even the amplification of sequences prone to secondary structures that lead to low PCR yields.

However, despite the comprehensive and well-done study to show that the designed superhelicase can be used in isothermal DNA amplification with applications similar to those of PCR, in my opinion the contribution is quite technical, without comparison with commercial tHDA (only compared to PCR). The scientific advance described in this work does not represent a significant step in relation to that described in a previous contribution of the authors, in which the bases are laid to obtain the superhelicase proposed in this work. For this reason, in my opinion, the article in its current form deserves publication in a more specialized journal.

Reviewer #2 (Remarks to the Author):

In this manuscript, Gavrilov and colleagues describe an improvement of the helicase-dependent amplification (HDA) through the use of an engineered DNA helicase, PcrA-M6, *G. stearothermophilus* PcrA helicase containing 6 mutations. Five of these mutations were introduced previously in PcrA to create a “super helicase” PcrA-X by trapping it in the unwinding competent conformation. Here, however, the authors suggest that cross-linking is not required for the mutant PcrA function in isothermal amplification. In addition to PcrA-M6, the SHARP reaction (the authors’ title for their version

of HAD) contains E.coli SSB, B. stearothermophilus-LF DNA polymerase, primers, template and dNTPs similar to the regular PCR, ATP, commercial pyrophosphatase and BSA.

The authors demonstrate that they can achieve efficient amplification of the templates up to 6,000 bp in length, which is remarkable for the isothermal amplification. They also show that the reaction can be carried out using a variety of templates including genomic and plasmid DNA.

Isothermal amplification can be very useful in many applications, especially when specialized equipment (i.e. thermocycler) cannot be used. SHARP technique seems to overcome many of the deficiencies of the previously described isothermal amplification methodologies.

As presented, however, the PcrA-M6 enzyme and what makes it suitable for SHARP is insufficiently described. It is unclear what parameters (processivity, reduced ATPase, increased force generation, ability to initiate unwinding without 3'-ssDNA overhang, etc.) is important for the M6 performance in SHARP?

The manuscript leaves it unclear why the cross-linking of PcrA-M5/M6 is not necessary to achieve an efficient unwinding. Previous work from the same lab (Arslan et al 2015 Science) showed that Rep-X helicase requires cross-linking for its enhanced activity and also left the impression that the same is true for PcrA-X; though after careful reading the latter is not claimed, just inferred. Without careful analysis of the structure/activity relationship in PcrA-M5/M6, this current manuscript does not rise to the rigor and impact expectation typically expected of Nature Communications paper.

Also, what was the logic for selecting the specific additional mutation in the PcrA-M6? Are all mutations carried out from the M5 version needed for activity?

Minor points:

All the components of the SHARP reaction come from the thermophilic sources except SSB and BSA. Does thermal stability of these proteins affect the reactions?

Fig.6. would greatly benefit from presenting the unwinding curves as FRET vs. time starting from the point where the reaction was initiated. The data in this figure also need to be quantified in terms of the unwinding rates and extents.

What is the fidelity of Bsu-LF DNA polymerase? How often the mutations are introduced when a plasmid sized DNA is amplified?

Reviewer #3 (Remarks to the Author):

The work reported is of a very high standard, presenting a new approach for the isothermal, helices-mediated amplification of large amplicons. The work is of significance and has potential application in multiple fields. The state of the art is clearly explained and the reported work significantly contributes to improving upon existing techniques. The work carried out and the results obtained support the conclusions and claims. The data analysis is very clear - all the relevant experiments are included as well as control experiments. The methodology is sound and adequate detail is provided to allow the work to be repeated in other laboratories. Overall, a high quality manuscript, with important results that will find wide potential impact. I recommend to publish as is.

REVIEWER COMMENTS

Reviewer #1 (Remarks to the Author):

This manuscript describes the application of an engineered superhelicase to develop an isothermal DNA amplification method, which authors named SHARP. This amplification method is similar to a commercially available one, Helicase Dependent Amplification (HDA), as the authors acknowledge. The main novelty is the use of a different helicase, a mutated one with additional modifications compared to a Superhelicase already described by the authors in reference 17. The Superhelicase is characterized by the ability to progressively unwind thousands of base pairs of DNA, with the final result of amplifying DNA sequences up to 6000 bp. HDA, which uses another helicase with limited unwinding processivity, is recommended for the amplification of short sequences (<150 nt). Within this framework, this is a significant contribution, demonstrating the higher processivity of the engineered helicase to develop an isothermal amplification DNA assay with performance comparable to PCR but at a constant temperature (optimal 65 °C). The authors demonstrated multiple applications for SHARP, including the amplification of linear DNA templates up to 6000 bp, the amplification of genomic DNA (474 amplicons), the quantification of CRISPR-Cas9 genome editing, the production of functional DNA that can be propagated in *E. coli* and even the amplification of sequences prone to secondary structures that lead to low PCR yields.

However, despite the comprehensive and well-done study to show that the designed superhelicase can be used in isothermal DNA amplification with applications similar to those of PCR, in my opinion the contribution is quite technical, without comparison with commercial tHDA (only compared to PCR). The scientific advance described in this work does not represent a significant step in relation to that described in a previous contribution of the authors, in which the bases are laid to obtain the superhelicase proposed in this work. For this reason, in my opinion, the article in its current form deserves publication in a more specialized journal.

We thank Reviewer #1 for comments and for suggesting comparing SHARP and tHDA, as this is a very important point. We directly compared SHARP with the golden-standard PCR in Figs. 2b and d for the 1463 bp amplicon at different template concentrations. We also compared SHARP with PCR for 474 bp genomic DNA sequence in Fig. 4. Although comparing SHARP and PCR with the commercial tHDA (helicase-dependent amplification) may seem like a reasonable request, in practice the comparison is not possible because tHDA cannot generate 474 or 1463 bp amplicons. tHDA is limited to only <150 bp amplicons [1,12,15]. Therefore, for 474 region of genomic DNA or for 1463 bp amplicons, tHDA would generate no product that can be gel imaged, further analyzed, quantified, or compared with SHARP. In addition, we are not claiming that SHARP is better than HDA/tHDA for the length of DNA that can be amplified by HDA/tHDA.

We included the following sentence to the paragraph.

“We note that HDA and tHDA are limited to only <150 bp amplicons”.

Reviewer #2 (Remarks to the Author):

In this manuscript, Gavrilov and colleagues describe an improvement of the helicase-dependent

amplification (HDA) through the use of an engineered DNA helicase, PcrA-M6, *G. stearothermophilus* PcrA helicase containing 6 mutations. Five of these mutations were introduced previously in PcrA to create a “super helicase” PcrA-X by trapping it in the unwinding competent conformation. Here, however, the authors suggest that cross-linking is not required for the mutant PcrA function in isothermal amplification. In addition to PcrA-M6, the SHARP reaction (the authors’ title for their version of HAD) contains *E. coli* SSB, *B. stearothermophilus*-LF DNA polymerase, primers, template and dNTPs similar to the regular PCR, ATP, commercial pyrophosphatase and BSA.

The authors demonstrate that they can achieve efficient amplification of the templates up to 6,000 bp in length, which is remarkable for the isothermal amplification. They also show that the reaction can be carried out using a variety of templates including genomic and plasmid DNA.

We thank reviewer for referring to our isothermal amplification as “remarkable”.

Isothermal amplification can be very useful in many applications, especially when specialized equipment (i.e. thermocycler) cannot be used. SHARP technique seems to overcome many of the deficiencies of the previously described isothermal amplification methodologies.

As presented, however, the PcrA-M6 enzyme and what makes it suitable for SHARP is insufficiently described. It is unclear what parameters (processivity, reduced ATPase, increased force generation, ability to initiate unwinding without 3'-ssDNA overhang, etc.) is important for the M6 performance in SHARP?

We thank Reviewer for this important suggestion. We find that 4 properties of PcrA are very important in enabling SHARP:

1. High processivity of PcrA M5/M6 to generate up to 6 kbp amplicons
2. The ability of PcrA M5/M6 to initiate unwinding without 3'-ssDNA overhangs in the presence of SSB. PcrA M5/M6 on its own (without SSB) is extremely inefficient at unwinding blunt ends.
3. Reduced PcrA M6/M5 ATPase activity keeps ATP abundant even during prologued amplification times characteristic to reactions with low copy numbers of the initial template. We reported previously that mutations made to create PcrA M5 reduce the ATPase activity [17].
4. We also add to this list thermostability and the ability of PcrA M5/M6 to remain active in a wide temperature range. This is important for selecting and using primers of different annealing temperatures.

The increased force resistance achieved by crosslinking may be less important for SHARP because we do not expect PcrA M5/M6 to encounter any resistance or roadblocks during unwinding in SHARP.

We summarize our response in the following paragraph added to the revision:

“Finally, we summarize what PcrA M6 properties are important for enabling SHARP. PcrA monomers are very good ssDNA translocases yet are ineffective as helicases in vitro [28]. With disulfide crosslinking, PcrA M5 and M6 become very effective helicase capable of unwinding several thousand base pairs after single binding to dsDNA. This property of the engineered helicase is important for generating kilobase-long amplicons in SHARP. PcrA plays an important role in plasmid replication and can unwind circular plasmids in the presence of the initiator protein RepD [32]; however, PcrA M5/M6 on its own requires a DNA substrate with 3'-ssDNA overhangs to initiate the unwinding [28, 17], making it unsuitable for SHARP. We have shown that in the presence of SSB, PcrA M5/M6 can effectively

unwind DNA with blunt ends and circular plasmids, and these abilities are likely to be essential for initiating each replication cycle in SHARP. PcrA M5 has reduced ATPase activity relative to PcrA [17] and this property keeps ATP abundant even during prologued amplification times characteristic to reactions with low copy numbers of the initial template. The last important property to mention is the thermostability of PcrA M5/M6 and their ability to remain active in a very wide temperature range. This enables many future applications in molecular diagnostics, where adjusting for the annealing temperature of primers or probes improves the diagnostics specificity.”

The manuscript leaves it unclear why the cross-linking of PcrA-M5/M6 is not necessary to achieve an efficient unwinding. Previous work from the same lab (Arslan et al 2015 Science) showed that Rep-X helicase requires cross-linking for its enhanced activity and also left the impression that the same is true for PcrA-X; though after careful reading the latter is not claimed, just inferred. Without careful analysis of the structure/activity relationship in PcrA-M5/M6, this current manuscript does not rise to the rigor and impact expectation typically expected of Nature Communications paper.

We thank Reviewer #2 for this point. We agree this needed further structure/activity analysis and clarification. In fact, the reviewer’s comment motivated us to re-examine one of our assumptions, that is, without an external crosslinker, the helicase is not constrained to the unwinding active conformation. This assumption turned to be wrong. We found that the ability to support DNA amplification is abolished if we add DTT to the reaction (and this was not the case if the helicase was crosslinked into the unwinding active conformation using an externally supplied crosslinker), strongly indicating that the two cysteines form spontaneous disulfide bonds, effectively creating a superhelicase.

Therefore, confining PcrA-M5/M6 to its active or closed conformation is still essential for the increased “superhelicase” activity in SHARP but applying an external crosslinker such as BMOE or BM(PEG)2 to M5/M6 is not necessary. When PcrA is in its closed conformation, the distance between CG atoms of N187 and L409 is approximately 4 Å according to the structure in Ref. [29]. Such a distance is close enough, so that after mutating two residues to cysteines (N187C and L409C) the disulfide bond between introduced C187 and C409 can form and keep M5 in its active or closed conformation without an external crosslinker.

We tested SHARP in the presence of DTT, because DTT can break the disulfide bond in PcrA M6. We found that DTT inhibits SHARP because the superhelicase activity of PcrA M5/M6 vanishes in the absence of the disulfide bond.

We included the following clarification: “PcrA M5 was previously [17] constrained to its closed conformation with an external crosslinker such as bismaleimidoethane (BMOE) or 1,8-bismaleimido-diethyleneglycol (BM(PEG)2). Here, we found that applying crosslinkers to M5 is not necessary. When PcrA is in its unwinding active, closed conformation, the distance between two residues N187 and L409 is approximately 4 Å [29]; hence, after mutating two residues to cysteines (N187C and L409C) a disulfide bond between C187 and C409 can form spontaneously and keep M5 in its active, closed conformation. In the Supplementary section, we further explored conditions for the disulfide bond formation in M5 and M6 mutants and tested how DTT affects SHARP. We found that an increased DTT concentration (>50 mM) inhibits SHARP, as expected because DTT breaks disulfide bonds essential for the superhelicase activity of our engineered enzyme in the absence of externally applied crosslinker (Fig. S2a). DTT had minimal effect on SHARP activities of cysteine-crosslinked helicase with BM(PEG)2 (Fig.

S2b). PcrA M5 at 400 nM showed higher unwinding activity (Fig. 6g) than PcrA, but the activity of M5 significantly decreased at 40 nM (Fig. 6h).”

Also, what was the logic for selecting the specific additional mutation in the PcrA-M6? Are all mutations carried out from the M5 version needed for activity?

Thank you very much for this great question. We chose to add one mutation (H93A) to PcrA M5 to create PcrA M6 because our previous study on a related helicase Rep showed that an equivalent mutation makes it 2-3 times faster as a single stranded DNA translocase. The figure below is from the thesis of Dr. Sinan Arslan, the first author of the Rep-X/PcrA-X paper. (Chapter 2, Figure 2.9).

Figure 2.9 Translocation speeds of biochemical Rep mutants. Normalized translocation speeds of mutants are plotted. Speeds were measured on smFRET repetitive shuttling assay and normalized by the speed of the baseline Rep (RepA333C, a single cysteine wild type like mutant (58)).

In a nanopore assay where PcrA M5/M6 unwinds and drives the DNA through MspA nanopore, we directly compared the unwinding speeds of PcrA M5 and PcrA M6 and found that H93A mutation in M6 increases the speed up to 4 times. We have further explored this result with molecular dynamics simulations and found that the mutation H93A likely lowers the energy barrier and reduces the friction for DNA translocation, leading to a higher unwinding activity.

We added the nanopore unwinding data and MD simulations to the revised manuscript, and added the following paragraphs:

“For better understanding of the underlying mechanistic properties of PcrA M5/M6, we employed Single-molecule Picometer Resolution Nanopore Tweezers (SPRNT) [31] and molecular dynamic (MD) simulations. SPRNT assay shows our engineered superhelicase can unwind several thousands of base pairs after single binding to a DNA substrate and we find this property important for generating kilo-bp-long amplicons. SPRNT assay also directly measures the helicase unwinding speed and shows that the introduced mutation H93A increases the helicase speed more than 3 times. This is consistent with the increased activity of PcrA M6 observed in the bulk FRET assay.

The SPRNT setup (Fig. 7e) consists of two wells separated by a phospholipid bilayer and connected with MspA nanopore inserted into the bilayer [31]. The well near the positive electrode contains the signal buffer (500 mM KCl and 50 mM HEPES pH 8.0), while the well near the negative electrode contains the reaction buffer (200 mM KCl, 50 mM HEPES pH 8.0, 5 mM MgCl₂, and 2 mM ATP) and helicase-DNA construct [31] at room temperature (22 °C). The DNA construct has 64 repeats of spinach DNA sequence where each repeat is 103 bp long. We apply 180 mV potential difference between the electrodes and measure pA current. As the helicase unwinds the DNA and drives one strand into the nanopore, the electric current passing through the nanopore generates a repetitive signal, Fig 7f. By counting number of 103-bp-long repeats in time, we can determine helicase unwinding speed. For PcrA M5 in Fig 7f, we observe 3.2 repeats during 8 seconds. For M6 we identified 10.4 repeats during 8 seconds. Figure S2 c and d show PcrA M5 and M6 unwinding ≈ 1500 and ≈ 3800 nucleotides after single binding to DNA. By analyzing unwinding of $\approx 10^4$ nucleotides, we find the average speed of PcrA M5 and M6 of 40 bp/s and 134 bp/s, respectively and conclude that at room temperature, mutation H93A increases the unwinding speed up to 4 times.

We used MD simulations for the PcrA-DNA complex to further investigate what role H93 residue plays in PcrA translocation. We observed that H93 served as a gating residue to control ssDNA translocation. As demonstrated in Fig. 8a, initially H93 interacted with the outgoing nucleobase (Nt_a). Through the course of the simulation, H93 disengaged from Nt_a and eventually formed interaction with the next departing nucleobase (Nt_b) in the final state (Fig. 8b). Next, we carried out simulations with H93 mutated into alanine. H93A did not form close contact with Nt_b in the final state, showing a diminished role in gating (Fig. 8d). We measured the contact area between Nt_b and H93/H93A in the final state (Fig. 8c). The average contact area for Nt_b - H93 is 32 Å², much larger than that for Nt_b - H93A (5 Å²). Similarly in the initial state, the average contact area for Nt_a - H93 is 40 Å², much larger than the 25 Å² measured for Nt_a - H93A (SI Fig.S-MD c). We thus suggest that the mutation H93A lowers the energy barrier and reduces the friction for DNA translocation, leading to a higher unwinding activity as was also observed in both bulk unwinding and SPRNT assays.”

Minor points:

All the components of the SHARP reaction come from the thermophilic sources except SSB and BSA. Does thermal stability of these proteins affects the reactions?

We included a brief paragraph to address this remark:

“Bovine Serum Albumin (BSA) and E. coli SSB used with SHARP at 65°C are not extracted from thermophilic organisms, but they both support SHARP at elevated temperatures, suggesting they maintain their supporting roles. BSA is frequently used in PCR mixes and can withstand temperatures above 65°C while early studies of E. coli SSB showed it remains stable and active after boiling and exposure to high temperatures [27].”

Fig.6. would greatly benefit from presenting the unwinding curves as FRET vs. time starting from the point where the reaction was initiated. The data in this figure also need to be quantified in terms of the unwinding rates and extents.

We agree. Figure 6 now shows FRET efficiency versus time starting from the point where the reaction was initiated by adding ATP. We obtained and quantified the unwinding efficiency for each helicase from the fit to FRET efficiency and summarized the rates in Fig 6j. Since the helicase concentration is varied for each FRET plot, we determined the relative activity by dividing the unwinding rate by the helicase concentration in bulk. Figure 6k shows helicase activity relative to Rep-X.

We also modified the text to comment on the unwinding rates and activities:

“Figure 6j summarizes the apparent unwinding rates of different helicases we tested. We also calculated normalized helicase activity by dividing the unwinding rate by the helicase concentration used (Fig. 6k inset), and plotted the normalized activity relative to Rep-X (Fig. 6k). Rep-X shows the highest unwinding activity, followed by UvrD, PcrA M6, PcrA M5, and PcrA in that order. Although PcrA M5/M6 are not as active as Rep-X or UvrD according to the FRET assay, only they supported SHARP.”

What is the fidelity of Bsu-LF DNA polymerase? How often the mutations are introduced when a plasmid sized DNA is amplified?

We believe this refers to Bst-LF DNA polymerase active at 65°C and used for majority of our assays. Bsu-LF was only used to demonstrate SHARP at 37°C. We included following sentences:

“For applications in cloning and gene manipulation, the amplification fidelity plays an important role because it determines how accurate the DNAP replicates the desired template. The error rate of Bst-LF used in SHARP was estimated to be about 60×10^{-6} errors/base [22], where 89-92 % of all errors are substitutions, 7-8% are deletion, and 1-3% insertions. Bst-LF is less prone to errors than Taq DNAP (2×10^{-4} to 2×10^{-5} errors/base), but more prone to errors than Phusion [23].

Reviewer #3 (Remarks to the Author):

The work reported is of a very high standard, presenting a new approach for the isothermal, helicase-mediated amplification of large amplicons. The work is of significance and has potential application in multiple fields. The state of the art is clearly explained and the reported work significantly contributes to improving upon existing techniques. The work carried out and the results obtained support the conclusions and claims. The data analysis is very clear - all the relevant experiments are included as well as control experiments. The methodology is sound and adequate detail is provided to allow the work to be repeated in other laboratories. Overall, a high quality manuscript, with important results that will

find wide potential impact. I recommend to publish as is.

We thank Reviewer #3 for their kind words about our manuscript and for referring to it as “a high quality manuscript”.

We introduced several minor changes:

1. “primers at 250 μ M” is changed to “primers at 250 nM”. The initially submitted manuscript had a typo.
2. Title is changed from “Engineered helicase eliminates thermal cycling from DNA amplification while retaining desired PCR characteristics” to “Engineered helicase replaces thermocycler in DNA amplification while retaining desired PCR characteristics”
3. “HDA or helicase-dependant amplification [12, 13] is very similar to SHARP” is replaced with “HDA or helicase-dependent amplification [12, 13] is very similar to our amplification method”, because the abbreviation SHARP is introduced later in the text.
4. In the Introduction paragraph, we added a sentence: “Here, we show that an engineered helicase can replace a thermocycler in DNA amplification while retaining the versatility and other desired characteristics of PCR. We also explore mechanistic properties of the engineered helicase with Single-molecule Picometer Resolution Nanopore Tweezers (SPRNT) and molecular dynamic simulations, and we show that enhanced processivity and speed of the engineered PcrA M6 helicase facilitate isothermal amplification.”
5. Other minor editing is marked in the text

REVIEWERS' COMMENTS

Reviewer #2 (Remarks to the Author):

The authors have adequately addressed my original comments. The addition of nanopore data and MD, as well as clarifications in the text significantly strengthen the paper.